# Efficient RNA drug delivery using red blood cell extracellular vesicles

Waqas Muhammad Usman[1], Tin Chanh Pham[1], Yuk Yan Kwok[2], Luyen Tien Vu[1], Victor Ma[2], Boya Peng[1], Yuen San Chan[1], Likun Wei[1], Siew Mei Chin[1], Ajijur Azad[1], Alex Bai-Liang He[3], Anskar Y.H. Leung[3], Mengsu Yang[1,4], Ng Shyh-Chang[5], William C. Cho[2], Jiahai Shi[1,6] & Minh T.N. Le [1,6]

Most of the current methods for programmable RNA drug therapies are unsuitable for the clinic due to low uptake efficiency and high cytotoxicity. Extracellular vesicles (EVs) could solve these problems because they represent a natural mode of intercellular communication. However, current cellular sources for EV production are limited in availability and safety in terms of horizontal gene transfer. One potentially ideal source could be human red blood cells (RBCs). Group O-RBCs can be used as universal donors for large-scale EV production since they are readily available in blood banks and they are devoid of DNA. Here, we describe and validate a new strategy to generate large-scale amounts of RBC-derived EVs for the delivery of RNA drugs, including antisense oligonucleotides, *Cas9* mRNA, and guide RNAs. RNA drug delivery with RBCEVs shows highly robust microRNA inhibition and CRISPR–Cas9 genome editing in both human cells and xenograft mouse models, with no observable cytotoxicity.

[1] Department of Biomedical Sciences, College of Veterinary Medicine and Life Sciences, City University of Hong Kong, 83 Tat Chee Avenue, Kowloon, Hong Kong. [2] Department of Clinical Oncology, Queen Elizabeth Hospital, 30 Gascoigne Road, Kowloon, Hong Kong. [3] Queen Mary Hospital and Department of Medicine, Li Ka Shing Faculty of Medicine, The University of Hong Kong, 102 Pok Fu Lam Road, Hong Kong Island, Hong Kong. [4] Key Laboratory of Biochip Technology, Biotech and Health Centre, City University of Hong Kong Shenzhen Research Institute, Shenzhen, China. [5] Genome Institute of Singapore, 60 Biopolis Street, 138672 Singapore, Singapore. [6] City University of Hong Kong Shenzhen Research Institute, Shenzhen, China. Correspondence and requests for materials should be addressed to M.T.N.L. (email: mle.bms@cityu.edu.hk)

RNA therapeutics including small-interfering RNAs (siR-NAs), antisense oligonucleotides (ASOs), and CRISPR–Cas9 genome editing guide RNAs (gRNAs) are emerging modalities for programmable therapies that target the diseased human genome with high specificity and great flexibility[1]. Although some chemically modified ASOs and siRNAs have reached clinical trials, they are still mostly limited to the liver and central nervous system due to the inherent targeting biases of current delivery vehicles[2,3]. Common vehicles for RNA drug delivery, including viruses (e.g., adenoviruses, lentiviruses, retroviruses), lipid transfection reagents, and lipid nanoparticles, are usually immunogenic and/or cytotoxic[4,5]. Thus a safe and effective strategy for the delivery of RNA drugs to most primary tissues and cancer cells, including leukemia cells and solid tumor cells, remains elusive[1,3]. Here we sought to harness eukaryotes' natural mechanism for RNA exchange and intercellular communication, the extracellular vesicles (EVs), to employ them as RNA drug delivery vehicles[6]. The natural delivery of microRNAs and mRNAs by EVs was first discovered in mast cells by Valadi et al.[7]. Subsequently, this phenomenon was also observed in many other cell types as an essential mode of intercellular signaling[8,9]. The natural biocompatibility of EVs with mammalian cells suggests that it can overcome most cellular barriers and drug delivery hurdles, such as RNase susceptibility, endosomal accumulation, phagocytosis, multidrug resistance, cytotoxicity, and immunogenicity[10,11]. Recent studies have successfully developed electroporation methods for loading siRNAs into EVs leading to robust gene silencing without any toxicity in neurons, cancer cells, and blood cells, suggesting that EVs are a new generation of drug carriers that enable the development of safe and effective gene therapies[11–13]. However, EV-based drug delivery methods are still in their infancy due to the limitations in EV production[14]. To produce highly pure and homogenous EVs, we need stringent purification methods such as sucrose density gradient ultracentrifugation or size exclusion chromatography but they are time-consuming and not scalable[14]. Moreover the yield is so low that billions of cells are needed to get sufficient EVs, and such numbers of primary cells are usually not available[14]. If immortalized cells are used to derive EVs instead, we run the risk of transferring oncogenic DNA and retrotransposon elements along with the RNA drugs[15]. In fact, all nucleated cells present some level of risk for horizontal gene transfer, because it is not predictable a priori which cells already harbor dangerous DNA, and which do not.

Thus we used human RBCs to produce EVs for RNA therapies because (i) RBCs lack both nuclear and mitochondrial DNA[16], (ii) RBCs are the most abundant cell type (84% of all cells) in the body[17]; and (iii) RBCs can be obtained from any human subject readily, and have been used safely and routinely for blood transfusions over decades[16]. In this study, we scaled up the generation of large amounts of RBCEVs for the delivery of therapeutic RNAs. RBCEV-mediated RNA drug delivery led to efficient microRNA knockdown and gene knockout with CRISPR–Cas9 genome editing in leukemia and breast cancer cells in vitro and in vivo, without any observable cytotoxicity. As RBCs are enucleated cells devoid of DNA, RBCEVs will not pose any risk of horizontal gene transfer. This study demonstrates a simple and efficient platform for RNA drug delivery that is safe, scalable, and applicable to any gene therapy.

## Results

### Purification and characterization of RBCEVs.
We have devised a new strategy to purify large-scale amounts of EVs from RBCs at low cost. RBCs were obtained from group O blood of healthy donors and treated with calcium ionophore overnight. The purification of RBCEVs was optimized with sequential centrifugation steps including the removal of protein contamination using a 60% sucrose cushion that yielded a homogenous population of EVs with an average diameter of ~140 nm and a polydispersity index ~0.07, determined using a Nanosight particle analyzer and Zetasizer (Fig. 1a and Supplementary Fig. 1a, b). Each unit of RBCs, isolated from ~200 ml blood, yielded $5-10 \times 10^{13}$ EVs. These EVs were negatively charged with a Zeta potential of $-11.5$ mV on average (Supplementary Fig. 1c). The morphology of the EVs appeared heterogeneous under a transmission electron microscope (TEM), with a mixture of both small exosome-like and large microvesicle-like particles (Fig. 1b). Purified RBCEVs were enriched in EV markers (ALIX and TSG101) and hemoglobin A, the major RBC protein (Fig. 1c). In addition, RBCEVs were also enriched in Stomatin (STOM), a marker of RBCEVs, but completely lacked Calnexin (CANX), an endoplasmic reticulum marker which is common in many cell types but absent in RBCs (Fig. 1d)[18]. These data illustrated the identity and purity of RBCEVs; there was no contamination from other types of blood EVs. Moreover, the RBCEVs were stable after multiple freeze–thaw cycles. There was no aggregation or any significant change in the morphology, concentration, or size distribution of the EVs after 1–3 freeze–thaw cycles as determined by using both TEM and Nanosight particle analysis (Supplementary Fig. 2a, b).

### Delivery of ASOs to leukemia cells using RBCEVs.
RBCEVs were taken up by leukemia cells at high efficiencies with no observable toxicity. After 24 h of incubation with RBCEVs, Western blot analysis of leukemia MOLM13 cells showed a clear uptake of Hemoglobin A, which was absent in the untreated cells (Fig. 1e). MOLM13 cells became ~99% fluorescent positive after a 24 h incubation with fluorescence-labeled EVs that was observed by both immunostaining and FACS (Fig. 1f, g and Supplementary Fig. 3). This uptake was reduced by 60–70% when heparin was added together with RBCEVs to the cells, suggesting that RBCEV uptake was dependent on heparan sulfate proteoglycans (Fig. 1g, h). We optimized the electroporation of RBCEVs with Alexa Fluor® 647 labeled Dextran and obtained up to 93.6% fluorescent EVs at the voltage of 250 V (Supplementary Fig. 4).

Subsequently we electroporated RBCEVs with a FAM-labeled scrambled negative control ASOs (FAM-NC-ASOs) with 2′O-methyl modification at every nucleotide (Fig. 2a). To compare the density of unelectroporated and electroporated RBCEVs, we layered $3.3 \times 10^{12}$ RBCEVs on top of a 10–60% sucrose gradient and separated the EVs using ultracentrifugation at $150,000 \times g$ for 16 h (Supplementary Fig. 5a). Both unelectroporated and electroporated RBCEVs were concentrated in fraction 5–7, at the interphase of the 20 and 40% sucrose layers, with the density of ~1.12–1.16 g/cm³ consistent with a previous report[19] (Supplementary Fig. 5b). Unelectroporated and electroporated RBCEVs in fraction 6 exhibited the same size distribution (Supplementary Fig. 5c). However, the concentration of electroporated RBCEVs was lower than unelectroporated RBCEVs in fraction 5–7 (Supplementary Fig. 5b). A red pellet was found at the bottom of the gradients of the electroporated (but not the unelectroporated) RBCEVs suggesting that some of the electroporated EVs formed aggregates. FAM fluorescence was enriched in fraction 5 to 7 of RBCEVs electroporated with FAM-labeled ASOs, indicating that RBCEVs were loaded successfully with FAM ASOs (Fig. 2b). Unbound FAM ASOs in fraction 1 emitted a higher FAM intensity than the EV-enriched fractions, suggesting that only a portion of FAM ASOs were loaded into the RBCEVs (Fig. 2b).

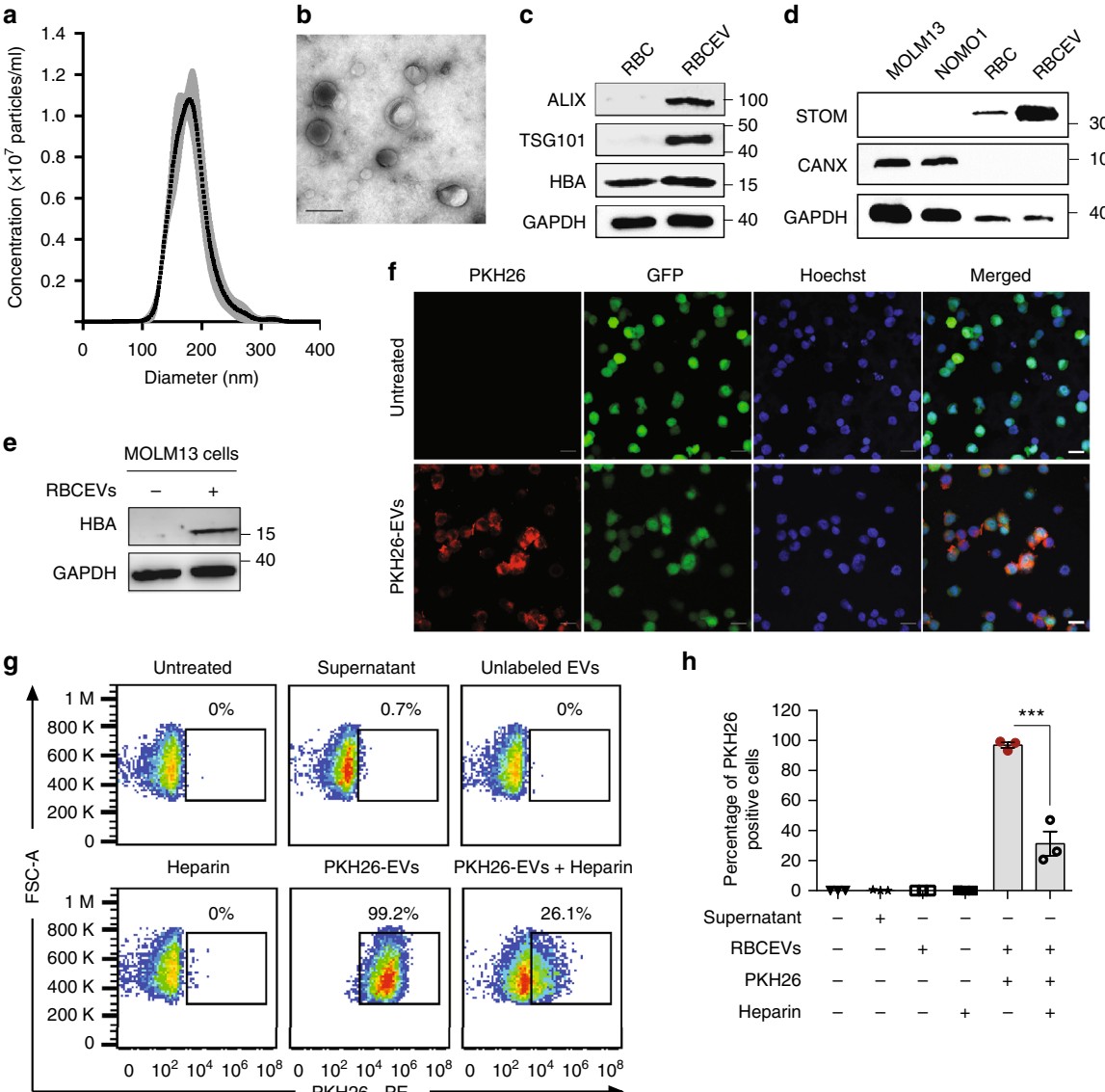

**Fig. 1** Characterization of EVs from RBCs and uptake of RBCEVs by leukemia cells. **a** Average concentrations (100,000 × dilution) of RBCEVs from three donors ± SEM (gray) and their size distribution, determined using a Nanosight nanoparticle analyzer. **b** Representative transmission electron microscopy image of RBCEVs. Scale bar: 100 nm. **c** Western blot analysis of EV markers ALIX and TSG101; and RBC marker Hemoglobin A (HBA) relative to GAPDH (loading control) in cell lysates and EVs from RBCs. **d** Western blot analysis of Stomatin (STOM) and Calnexin (CANX) as the markers of RBCEVs and endoplasmic reticulum, respectively, relative to GAPDH, in leukemia MOLM13 cells, NOMO1 cells, RBCs, and RBCEVs. **e** Western blot analysis of HBA relative to GAPDH in leukemia MOLM13 cells untreated or incubated with $8.25 \times 10^{11}$ RBCEVs for 24 h. **f** Representative immunofluorescent images of MOLM13-GFP cells incubated with $12.4 \times 10^{11}$ PKH26-labeled EVs for 24 h. Scale bar, 20 μm. **g** FACS analysis of PKH26 in MOLM13 cells that were incubated with $12.4 \times 10^{11}$ unlabeled or PKH26-labeled EVs with and without Heparin for 24 h. The supernatant of the last wash after PKH26 labeling was used to determine the background. Percentages of PKH26-positive cells are indicated above the gates. **h** Average percentage of PKH26-positive cells in each condition (mean ± SEM, $n = 3$ cell passages). P value (***$P < 0.001$) was determined using Student's one-tail t-test. In **c–e**, molecular weights (KDa) of protein markers are shown on the right. Each experiment was repeated two to three times in 2–3 cell passages

To improve our estimate of RNA loading into electroporated RBCEVs, we separated unbound ASOs from the electroporated RBCEVs using a 10% native gel and found that ~76% of the ASOs migrated into the gel from the $8.25 \times 10^{11}$ ASO-electroporated RBCEV sample relative to the total 200 pmol ASOs (Fig. 2c). Hence, ~24% of the ASOs were loaded into the RBCEVs by electroporation. Similar loading efficiencies were observed based on the FAM fluorescence in RBCEVs electroporated with FAM ASOs (Supplementary Fig. 6a). It is noteworthy that unbound ASOs appeared as a single band in all samples including the electroporated RBCEVs and no FAM signal was detected in the wells. Hence, the electroporation did not cause any aggregation of

the ASOs, unlike the aggregation of Cy3 or Cy5-labeled oligonucleotides that we and others observed before[20]. To test the RNA stability within RBCEVs in blood serum-like conditions, we incubated FAM-ASO-electroporated RBCEVs or an unelectroporated mixture of FAM ASOs and RBCEVs with 50% FBS, which contained various nucleases at 37 °C for 72 h. The fluorescent signal of electroporated FAM ASOs declined at a significantly lower rate than unelectroporated FAM ASOs (Supplementary Fig. 6b). This data suggested that the FAM ASOs were protected from extracellular nuclease-mediated degradation after incorporation into RBCEVs.

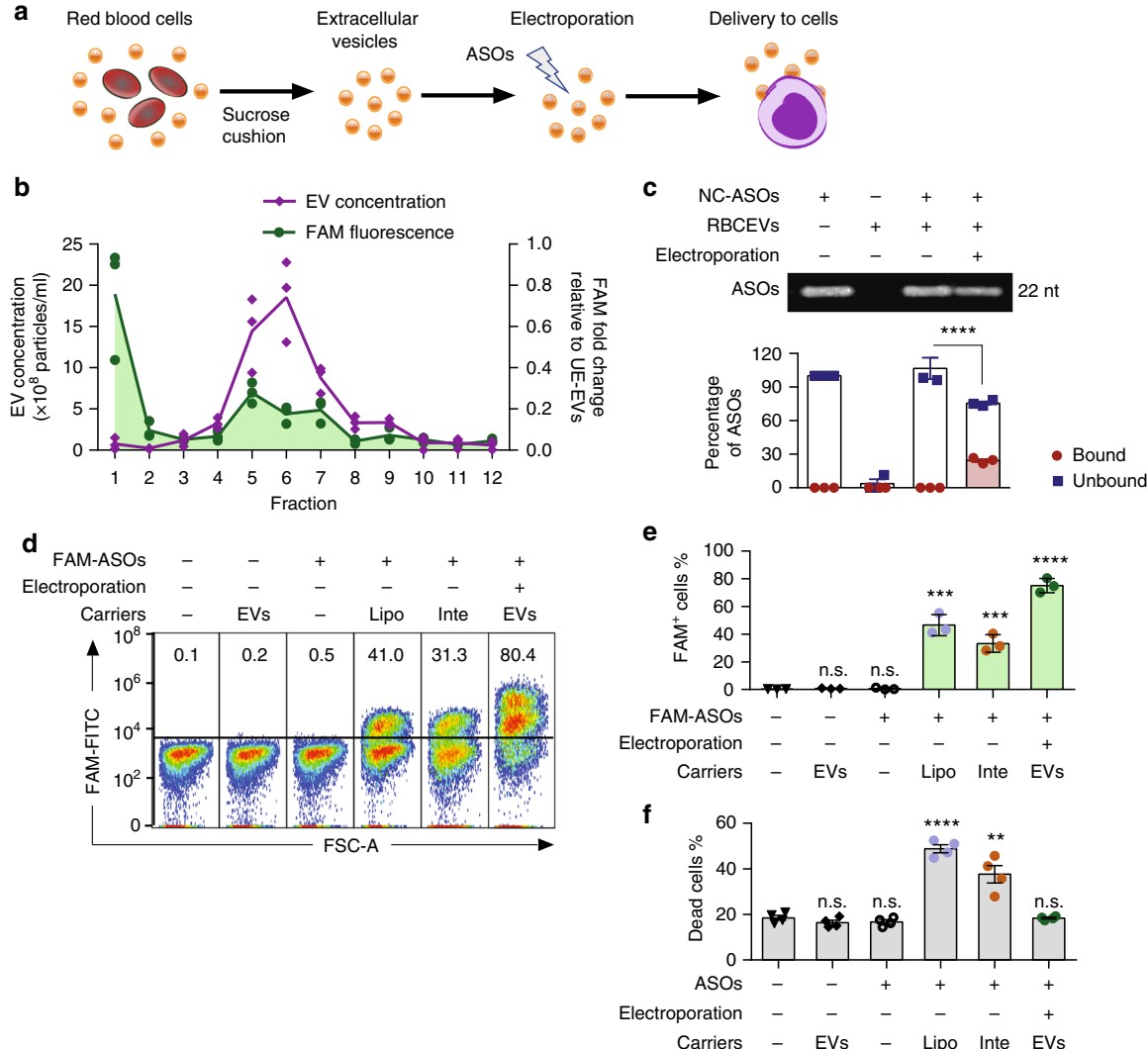

**Fig. 2** Electroporation of RBCEVs with ASOs and delivery to leukemia cells. **a** Experimental scheme of ASOs delivery by RBCEVs. **b** Average concentration of EVs (200× dilution) and average fold change in FAM fluorescent intensity relative to unelectroporated EVs (UE-EVs) of 12 fractions in a sucrose gradient separation of RBCEVs electroporated with a FAM-labeled scrambled negative control ASOs (FAM-NC-ASOs), determined using a Nanosight analyzer and Synergy fluorescent microplate reader, respectively, $n = 3$ repeats. **c** Separation of unbound NC-ASOs (unlabeled) from $8.25 \times 10^{11}$ unelectroporated or electroporated RBCEVs compared to the untreated NC-ASOs (200 pmol) in 10% native gel, visualized using SYBR Gold staining (top) and the average percentage of NC-ASOs unbound or bound to electroporated RBCEVs (bottom), $n = 3$ independent replicates. **d** FACS analysis of FAM fluorescence vs. forward scatter area (FSC-A) in MOLM13 cells transfected with 400 pmol FAM-NC-ASO using Lipofectamine™ 3000 (Lipo), INTERFERin® (Inte) or $12.4 \times 10^{11}$ RBCEVs. Percentages of FAM-positive cells are indicated above the gate. **e** Average percentages of FAM+ cells among viable MOLM13 cells transfected or treated with RBCEVs containing FAM ASOs as in **d**, $n = 3$ repeats. **f** Percentages of dead cells determined by propidium iodide staining among MOLM13 cells transfected or treated with RBCEVs containing unlabeled NC-ASOs, $n = 4$ repeats. All graphs present mean ± SEM. Student's one-tail $t$-test results are shown as n.s. non-significant; **$P < 0.01$; ***$P < 0.001$ and ****$P < 0.0001$ relative to the unelectroporated control (**c**) or to the untreated control (**e**, **f**)

We consistently observed 70–80% uptake of FAM-labeled ASOs and Alexa Fluor® 647 labeled Dextran in leukemia MOLM13 cells after 24-hour-treatments with ~$12 \times 10^{11}$ RBCEVs for $5 \times 10^4$ cells, which was the optimal dose (Fig. 2d and Supplementary Fig. 7). The uptake of FAM ASOs was ~88% corresponding to ~85% uptake of fluorescent-labeled RBCEVs in another leukemia cell line, NOMO1 (Supplementary Fig. 8a - c). FAM fluorescence was observed in ~0.4% MOLM13 and NOMO1 cells after 24 h of incubation with unencapsulated (unbound) FAM ASOs (Fig. 2d, e and Supplementary Fig. 8c). Over 4 days, this signal increased to 2.1% of MOML13 cells suggesting that the unencapsulated FAM ASOs were taken up slowly by a tiny population of MOLM13 cells via gymnotic delivery (Supplementary Fig. 9). After 4 days, the percentage of FAM-positive MOLM13 cells did not increase further with FAM-ASO treatment. However, during the same period of time, the delivery of FAM ASOs by RBCEVs occurred at a much higher rate in MOLM13 cells, from 75% after 2 days to 100% FAM-positive cells after 4 days of incubation (Supplementary Fig. 9). These data indicated that RBCEVs conferred remarkable delivery efficiencies, since leukemia cells and most blood cells are considered cell types that are very difficult to transfect. Commercial transfection reagents, including Lipofectamine™ and INTERFERin®, could only produce ~3% uptake of Dextran and 33–46% uptake of ASOs in MOLM13 cells after 24 h of transfection (Fig. 2d, e and Supplementary Fig. 10a). Moreover, RBCEVs did not cause any toxicity to the cells. This is in contrast to the ~20–30% increase in cell death caused by Lipofectamine™

3000 and INTERFERin® (Fig. 2f and Supplementary Fig. 10b). Therefore, RNA delivery by RBCEVs show higher efficiency and lower toxicity in leukemia cells, compared to current transfection vehicles.

**Inhibition of miR-125b using 125b-ASO-loaded RBCEVs.** We further investigated the therapeutic potential of RBCEVs in delivering ASOs that antagonizes miR-125b, a well-known oncogenic microRNA in leukemia cells, prostate cancer, and breast cancer[21–27]. In particular, miR-125b is an oncomiR in refractory cancers such as acute myeloid leukemia and chemoresistant breast tumors, both of which are difficult to treat[22,24,28]. We and others have shown that miR-125b promotes the survival of cancer cells by repressing multiple genes in the p53 tumor suppressor network[24,27,29–31]. These studies suggested that miR-125b is a potential drug target for cancer treatment. However, no effective therapy has been developed to target miR-125b yet. Here, we electroporated anti-miR-125b ASOs (125b-ASOs) into RBCEVs then quantified the loading of 125b-ASOs in the EVs and the delivery of the ASOs to MOLM13 cells (Fig. 3a). Treatment of $6.2 \times 10^{11}$ electroporated RBCEVs with RNase $I_f$ led to a degradation of ~80% 125b-ASOs, relative to the untreated ASOs, quantified using a sequence-specific Taqman qRT-PCR; whereas, the same amount of ASOs in an unelectroporated mixture with RBCEVs was completely degraded (Fig. 3b). This data suggests that approximately 20% of the ASOs (~$24 \times 10^{12}$ copies) were loaded into RBCEVs by electroporation and thus, protected from the RNase. To quantify the copy number of 125b-ASOs, we generated a standard curve of the ASOs amplification using Taqman qRT-PCR (Supplementary Fig. 6c). Based on this curve, we found ~$21 \times 10^9$ copies of 125b-ASOs in MOLM13 cells after a 72-h-incubation with $12 \times 10^{11}$ 125b-ASO-loaded RBCEVs (Fig. 3c).

With the uptake of 125b-ASOs, the level of miR-125b was suppressed by 80–95% in MOLM13 cells in a dose-dependent manner relative to U6b RNA quantified using Taqman qRT-PCR (Fig. 3d). This result was confirmed using miRCURY-LNA qRT-PCR with miR-103a as the internal control (Supplementary Fig. 15a). miR-125a, the homolog of miR-125b, was also suppressed by 50–80% in a dose-dependent manner (Supplementary Fig. 11a). The same effects were observed in NOMO1 cells (Supplementary Fig. 11b). The inhibition of the miR-125 family led to a significant increase in BAK1, a target of the miR-125 family that we previously identified (Fig. 3e)[30]. The ASOs alone did not have any effect on miR-125b or BAK1 expression (Fig. 3d, e). Treatment with 125b-ASO-loaded RBCEVs also significantly dampened the growth of MOLM13 cells after 3–4 days of incubation (Fig. 3f). To test the function of miR-125b in another cancer type, we applied the same treatment to human breast cancer MCF10CA1a (CA1a) cells. We found a significant knockdown of miR-125a/b and reduced survival of CA1a cells after treatment with 125b-ASO-loaded RBCEVs (Supplementary Fig. 11c and Fig. 3g).

**In vivo distribution of RBCEVs in a breast cancer model.** We tested the uptake and distribution of RBCEVs in vivo using the xenograft model of breast cancer CA1 cells, known to be very aggressive and metastatic[32]. Luciferase-expressing CA1a cells were implanted subcutaneously in the left and right flanks of female nude mice (Fig. 4a). After 1 week (as the tumors approached 7 mm in diameter), we injected the left tumors with PKH26-labeled RBCEVs. The fluorescence signal became concentrated in the tumors and gradually declined over time (Fig. 4b), observed using an in vivo imaging system (IVIS). After 72 h, the PKH26 signal was still detectable in the tumors but

undetectable in other parts of the body (Fig. 4c, d). High resolution images of the tumor sections confirmed the internalization of PKH26-labeled RBCEVs by cells in the tumors (Fig. 4e). By contrast, when we injected PKH26 or DiR-labeled EVs intraperitoneally (i.p.) into nude mice bearing flank tumors, PKH26, or DiR fluorescence was widely dispersed in the body (Supplementary Fig. 12a, b). DiR signal was enriched in the liver, spleen, stomach, intestine, kidneys, and lung (Supplementary Fig. 12b, c). In cryosections of tissues from PKH26-EV i.p. injected mice, we found some RBCEV uptake in the tumor after i.p. injection, but much less than intratumoral injection (Supplementary Fig. 12d). We did not observe any inflammation, nor changes in morphology and cellular contents of the liver and other organs after these injections (Supplementary Fig. 13). Hence, local injection delivered RBCEVs more effectively to the tumors while systemic administration distributed RBCEVs to multiple organs without any significant cytotoxicity in nude mice.

**Intratumoral injection of 125b-ASO-loaded RBCEVs suppresses breast cancer.** After validating our RBCEV platform's potential utility in vivo, we used it to target miR-125b, which has not been tested for its role in breast tumorigenesis in vivo. We delivered 125b-ASO-loaded RBCEVs into luciferase-labeled CA1a tumors by intratumoral injections every 3 days (Fig. 5a). Breast tumor growth was significantly dampened by 125b-ASO-loaded RBCEVs, as observed from the decrease in tumor bioluminescence compared to the NC-ASO-loaded RBCEVs after 30–42 days of treatment (Fig. 5b, d). Injection of 125b-ASOs without RBCEVs did not result in any significant change in tumor growth compared to the NC-ASOs treatment. There was no significant difference in overall weight between the controls and 125b-ASO-loaded-RBCEV treated mice, suggesting that 125b-ASO-loaded RBCEVs induced tumor shrinkage specifically, without causing overall weight loss and toxicity (Fig. 5c). When harvested, the 125b-ASO-loaded-RBCEV treated tumors were smaller than the controls (Fig. 5e). Hematoxylin and eosin (H & E) staining of tumor sections showed that 125b-ASO-loaded-RBCEV treated tumors were also less invasive, and less metastasis was observed in the lung (Fig. 5f). Remarkably, miR-125b was reduced by ~95% in the 125b-ASO-loaded-RBCEV treated tumors (Fig. 5g). These data suggested that RBCEVs were avidly taken up by breast cancer cells in vivo, and that RBCEVs can deliver therapeutic ASOs to effectively antagonize oncomiRs and suppress tumorigenesis without any observable side effects.

**Systemic injection of 125b-ASO-loaded RBCEVs suppresses AML progression.** We have shown that RBCEVs are robust vehicles that delivered 125b-ASOs readily to AML cells for effective inhibition of miR-125b function in vitro. To test the functional efficacy of 125b-ASO-loaded RBCEVs in vivo, we sought to establish a xenograft model of AML in NSG mice. First, we determined the distribution of RBCEVs in the circulation of these mice following systemic administration of the EVs (Fig. 6a). Immediately after an intravenous (i.v.) injection of $3.3 \times 10^{12}$ PKH26-labeled RBCEVs (hereafter, referred as one dose), we found more than 40% circulating EVs positive for PKH26 (Fig. 6b). The percentage of PKH26-positive EVs declined over 6 h and remained at 3–4.5% after 12 h. The decrease in circulating human RBCEVs suggested that some of the EVs were taken up by the mouse tissues over time.

To confirm that RBCEVs can be distributed to various organs of NSG mice by i.p. injection, we administered two doses of DiR-labeled RBCEVs i.p. 24 h apart. 24 h after the second dose, we found bright DiR signals in the liver, spleen, stomach, and intestine using the fluorescence IVIS (Fig. 6c - e). Robust delivery

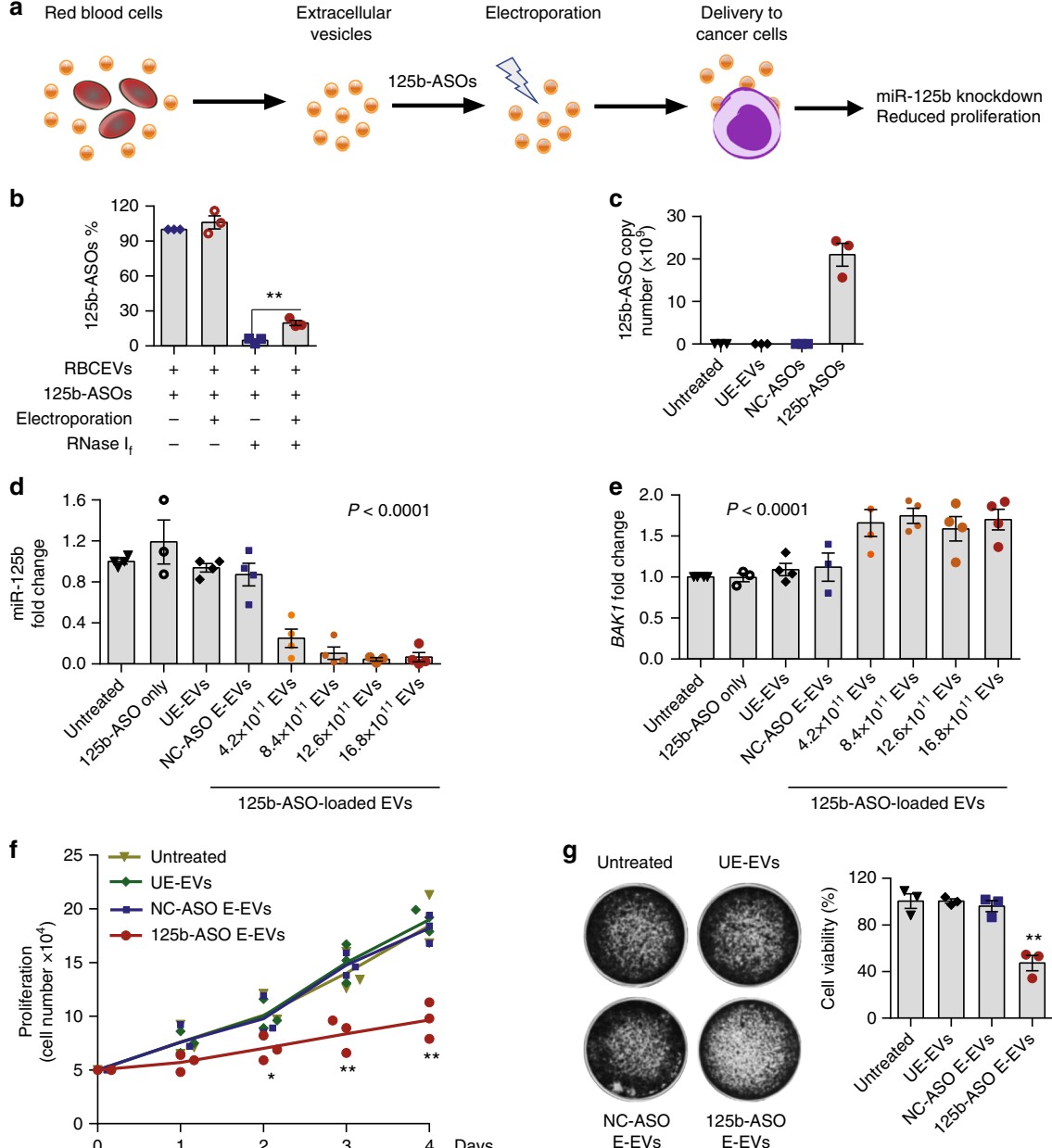

**Fig. 3** RBCEVs deliver ASOs to leukemia and breast cancer cells for miR-125b inhibition. **a** Experimental scheme of ASOs delivery to cancer cells using RBCEVs. **b** Percentage of anti-miR-125b ASOs (125b-ASOs) associated with $6.2 \times 10^{11}$ unelectroporated or 125b-ASO-electroporated RBCEVs after a treatment with RNase I$_f$ for 30 min. **c** Copy number of 125b-ASO in MOLM13 cells treated with $12.4 \times 10^{11}$ RBCEVs unelectroporated (UE-EVs) or RBCEVs electroporated with NC-ASOs or with 125b-ASOs for 72 h. **d** Expression fold change of miR-125b in MOLM13 cells that were incubated with 125b-ASOs alone, $16.8 \times 10^{11}$ unelectroporated RBCEVs (UE-EVs), $16.8 \times 10^{11}$ NC-ASOs-loaded RBCEVs, or 4.2 to $16.8 \times 10^{11}$ 125b-ASOs loaded RBCEVs. miR-125b expression was determined using Taqman qRT-PCR normalized to U6b RNA and presented as average fold change relative to the untreated control. **e** Expression fold change of *BAK1* in MOLM13 cells treated as in **d**, determined using SYBR Green qRT-PCR, normalized to *GAPDH* and presented as average fold change relative to the untreated control. **f** Proliferation of MOLM13 cells treated with $12.4 \times 10^{11}$ unelectroporated or NC/125b-ASO-electroporated EVs, determined using cell counts. **g** Viability of breast cancer CA1a cells (%) treated as in **f**, determined by crystal violet staining. In all panels, the experiments were repeated three or four times with 3 or 4 cell passages. Bar graphs present mean ± SEM. *P* values were calculated using one-way ANOVA test (**d**, **e**) or student's one-tail *t*-test relative to the untreated controls (**b**, **f**, **g**) *$P < 0.05$, **$P < 0.01$

of RBCEVs to the internal organs suggested that i.p. administered RBCEVs could effectively treat leukemia cells in the liver and spleen where leukemia usually develop. Due to the blocking of DiR signals by dense bone and the unavailability of microscope filters or cytometer filters with long excitation/emission wavelengths (750/780 nm for DiR), we were unable to detect DiR from the excised bone or bone marrow aspirates. Because bone marrow is the primary compartment that leukemia cells home into, we

attempted to determine the uptake of RBCEVs by bone marrow cells using RBCEVs labeled with Vivo-Track-680 (VVT), a near-infrared membrane dye that is detectable by FACS (Fig. 6f). Indeed, VVT fluorescence was detected in ~40% of the bone marrow cells from the NSG mice injected i.p. with VVT-labeled RBCEVs using FACS analysis (Fig. 6g, h). Hence, RBCEVs were robustly taken up by bone marrow cells and could deliver therapeutic molecules for leukemia treatment in vivo.

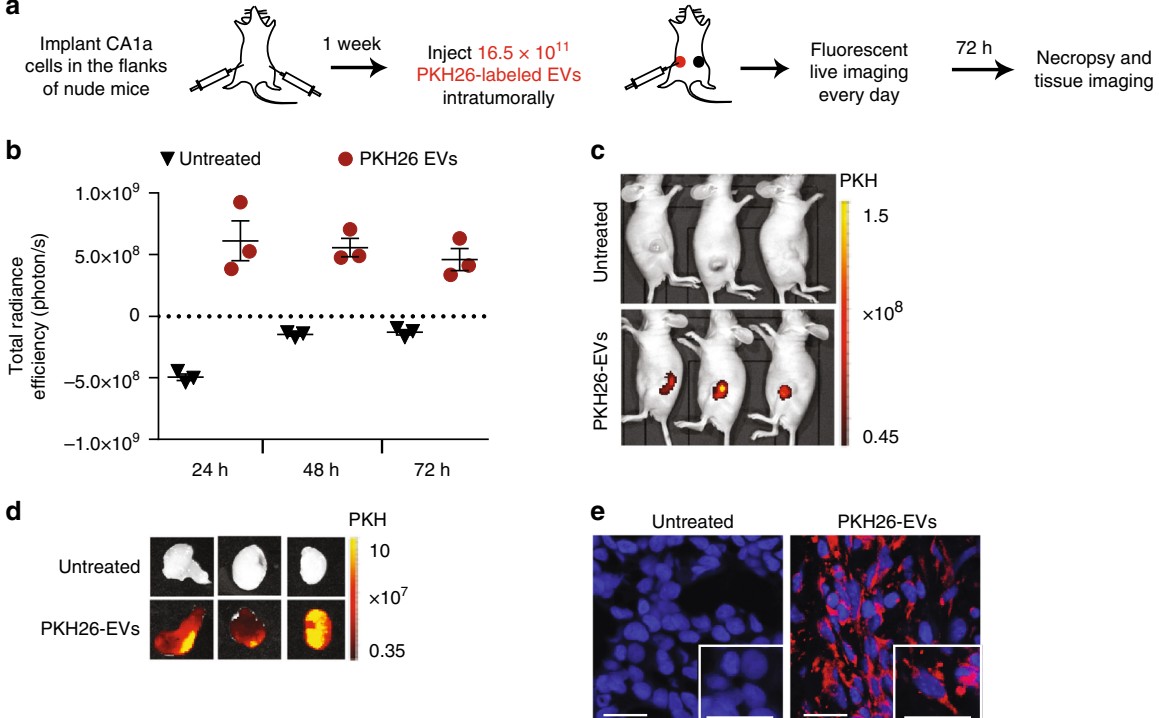

**Fig. 4** RBCEVs are taken up by breast cancer cells in vivo. **a** Schema of an in vivo EV uptake assay. **b** Total radiance efficiency of PKH26 fluorescence in the tumors 24 to 72 h after an intratumoral injection of 16.5 × 10¹¹ PKH26-labeled RBCEVs, determined using an in vivo imaging system (IVIS), presented as mean ± SEM ($n = 3$ mice). **c** Images of the mice bearing untreated tumors on the right flank and tumors injected with PKH26 -labeled EVs on the left flank, 72 h post-treatment, captured using IVIS. PKH26 is shown in pseudocolored radiance. **d** Images of the tumors excised from the mice in **c**. **e** Representative confocal microscopy images of tumor sections with DAPI stained nuclei and PKH26 signals from the cells with EV uptake. Scale bar, 20 μm

Subsequently, we generated AML xenografts by injecting luciferase-GFP-labeled MOLM13 cells in the tail vein of busulfan-conditioned NSG mice (Fig. 7a). After 1 week when leukemic bioluminescent signals became visible, we treated the mice with one dose of 125b-ASO-loaded RBCEVs every other day. The leukemia developed very rapidly so we could treat the mice for only 9 days before the control group paralyzed and died (usually 18–20 days after the cell inoculation). On day 9, the leukemic bioluminescence in mice treated with 125b-ASO-loaded RBCEVs decreased significantly compared to the control group (Fig. 7b). The control mice became very weak while the 125b-ASO-EVs treated mice were still active. The leukemic biolumi-nescent signals spread all over the mice' bodies, accumulating highly in the bone marrow, liver, and spleen of control mice, compared to the 125b-ASO-loaded-RBCEVs treated group (Fig. 7c). We could not assess the effect of the treatments on the overall survival of the mice due to restrictions defined by our institutional ethics committee. All the mice were killed on day 9 except for 2 control mice that died on day 8. GFP⁺ leukemia cells accounted for 63–70% cells in the bone marrow of the controls but reduced to 27–46% in the treated mice albeit there was no change in the body weight (Fig. 7d, e). H & E staining revealed extensive infiltration of leukemia cells in the liver and spleen of the control group while less leukemia cells were found in the 125b-ASO-loaded-RBCEV treated group (Fig. 7f). Moreover, qPCR analysis showed a significant knockdown of miR-125b in the spleen and liver (Fig. 7g). These data indicated that 125b-ASOs delivered by RBCEVs was taken up by leukemia cells and effectively suppressed leukemia progression in this model. Thus RNA inhibition using systemic administration of ASO-loaded RBCEVs may represent a new approach for leukemia treatment.

**Genome editing mediated by *Cas9* mRNA and gRNA-loaded RBCEVs.** Furthermore, we validated our RBCEV platform for gRNA-mediated genome editing with CRISPR–Cas9 (Fig. 8a). To test the feasibility of mRNA delivery using RBCEVs, we electro-porated HA-tagged *Cas9* mRNA (4,521 nucleotides) into RBCEVs, and used them to treat MOLM13 cells. We first quantified the loading of *Cas9* mRNA in RBCEVs using qPCR. Electroporated *Cas9* mRNA was protected by RBCEVs from RNase I$_f$ mediated degradation while unelectroporated *Cas9* mRNA was completely degraded (Fig. 8b). Specifically, about 18% of *Cas9* mRNA was loaded and protected in RBCEVs. We detected abundant *Cas9* mRNA in MOLM13 cells after a 24-h incubation of the cells with *Cas9* mRNA electroporated RBCEVs, whereas the cells treated with unelectroporated RBCEVs had no detectable *Cas9* mRNA (Fig. 8c and Supplementary Fig. 15b). *Cas9* protein was efficiently expressed in the nuclei of ~50% MOLM13 cells at 48 h post-treatment, detected using immu-nostaining with an anti-HA tag antibody (Fig. 8d, e). Western blot analysis confirmed the expression of Cas9 protein using an anti-Cas9 antibody (Fig. 8f).

Subsequently, we designed a gRNA targeting the human *mir-125b-2* locus with potential mutation site in the seed sequence of the miRNA (Supplementary Fig. 14a). This gRNA may bind to the other loci of the miR-125 family due to sequence similarity. Treatment of MOLM13 cells with RBCEVs loaded with *Cas9* mRNA and 125b-gRNA resulted in ~98% reduction of miR-125b expression and 90% reduction of miR-125a after a 2-day treatment (Fig. 8g and Supplementary Fig. 14b). As a conse-quence, *BAK1* was upregulated by approximately three fold (Fig. 8g). Sequencing data confirmed a cleavage site 3–8 nucleotides apart from the protospacer adjacent motif (PAM)

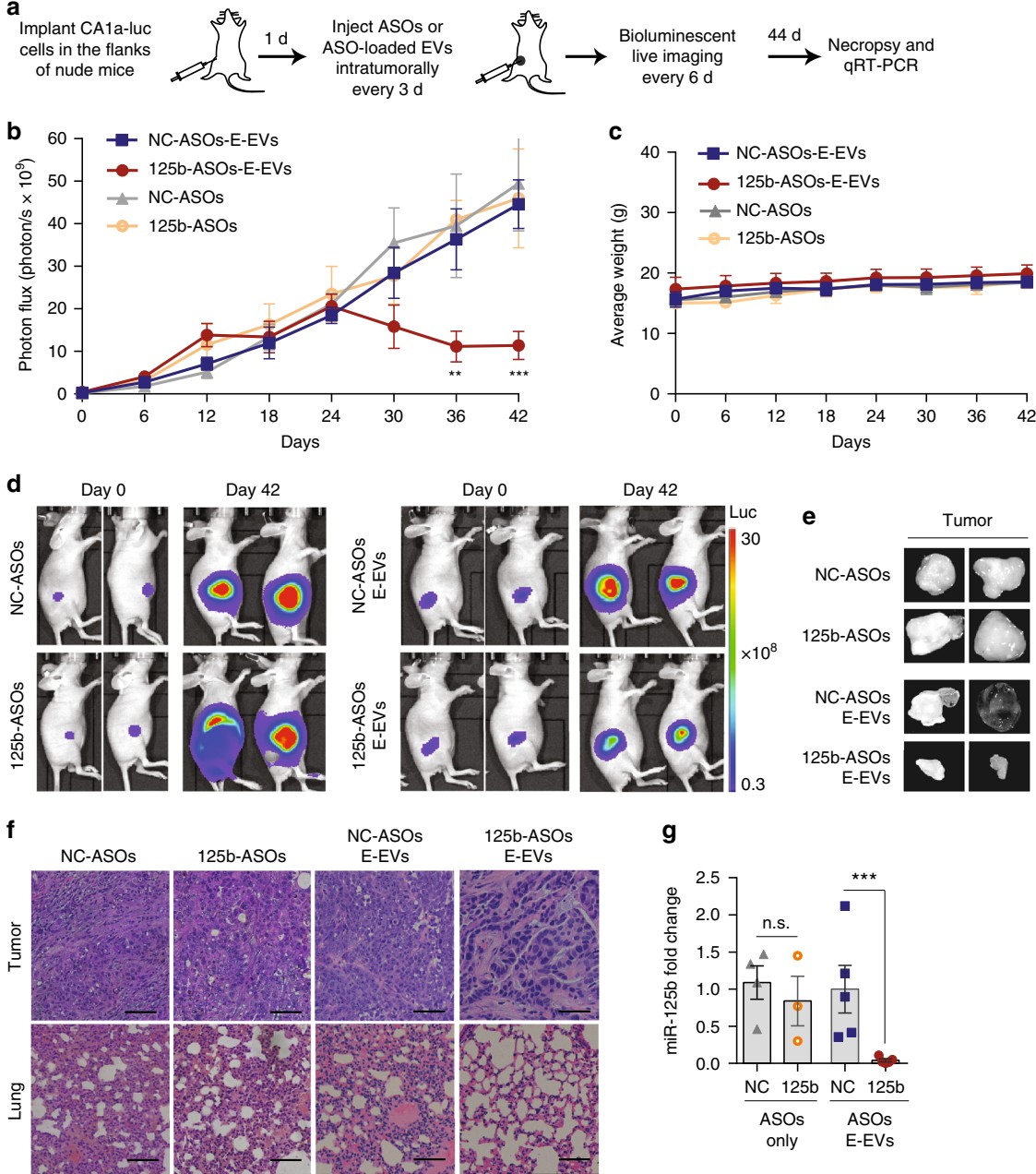

**Fig. 5** Treatment with ASOs-loaded RBCEVs suppresses tumor growth by miR-125b knockdown. **a** Schema of ASOs delivery to nude mice bearing breast cancer xenografts. **b** Average bioluminescent photon flux of the tumors treated every 3 days with intratumoral injection of $8.25 \times 10^{11}$ RBCEVs containing NC/125b-ASOs (E-EVs, $n = 8$ mice) or with 400 pmol NC/125b-ASOs ($n = 6$ mice), determined using IVIS (mean ± SEM). **c** Average weight of the mice (mean ± SEM). **d** Representative images on day 0 and 42. Bioluminescence is shown in pseudocolored radiance. **e** Representative pictures of the tumors on day 44. **f** Representative H & E staining images of the tumor and the lung collected on day 44. Scale bar, 50 μm. **g** miR-125b fold change relative to U6b RNA and NC condition in the tumors after 44 days of treatments, determined using Taqman qRT-PCR (mean ± SEM). P values were determined using one-tail Mann–Whitney test **b**, **g**: **$P < 0.01$; ***$P < 0.001$; n.s. non-significant. The whole experiment was performed in three independent repeats (three batches of mice)

sequence in each of the mutant clones (Fig. 8h). Insertions and deletions of different sizes that disrupted the mature miR-125 sequence were found at the cleavage sites (Fig. 8h). The rapid and high efficiency of miR-125a/b suppression was probably due to the short half-life of miR-125a/b, in addition to genome editing. These data suggest that RBCEVs are able to deliver a functional CRISPR–Cas9 genome editing system into leukemia cells effectively.

To test if RBCEVs can also deliver DNA plasmids, we electroporated two plasmids expressing *Cas9* and GFP gRNA into RBCEVs and treated 293T-eGFP cells for a week. EGFP knockout was observed in only ~10% cells (Supplementary Fig. 14c), likely due to the large sizes of the DNA plasmids. RBCEVs were also electroporated with a combination of *Cas9* mRNA and anti-eGFP gRNA at a 6:50 molar ratio. The *Cas9* mRNA/gRNA-loaded RBCEVs led to a complete loss of eGFP in ~32% NOMO1-eGFP cells (Supplementary Fig. 14d). Hence, RBCEVs can be used to deliver RNAs and DNAs for genome editing, albeit with lower efficiency for large DNA plasmids and higher efficiency for RNAs.

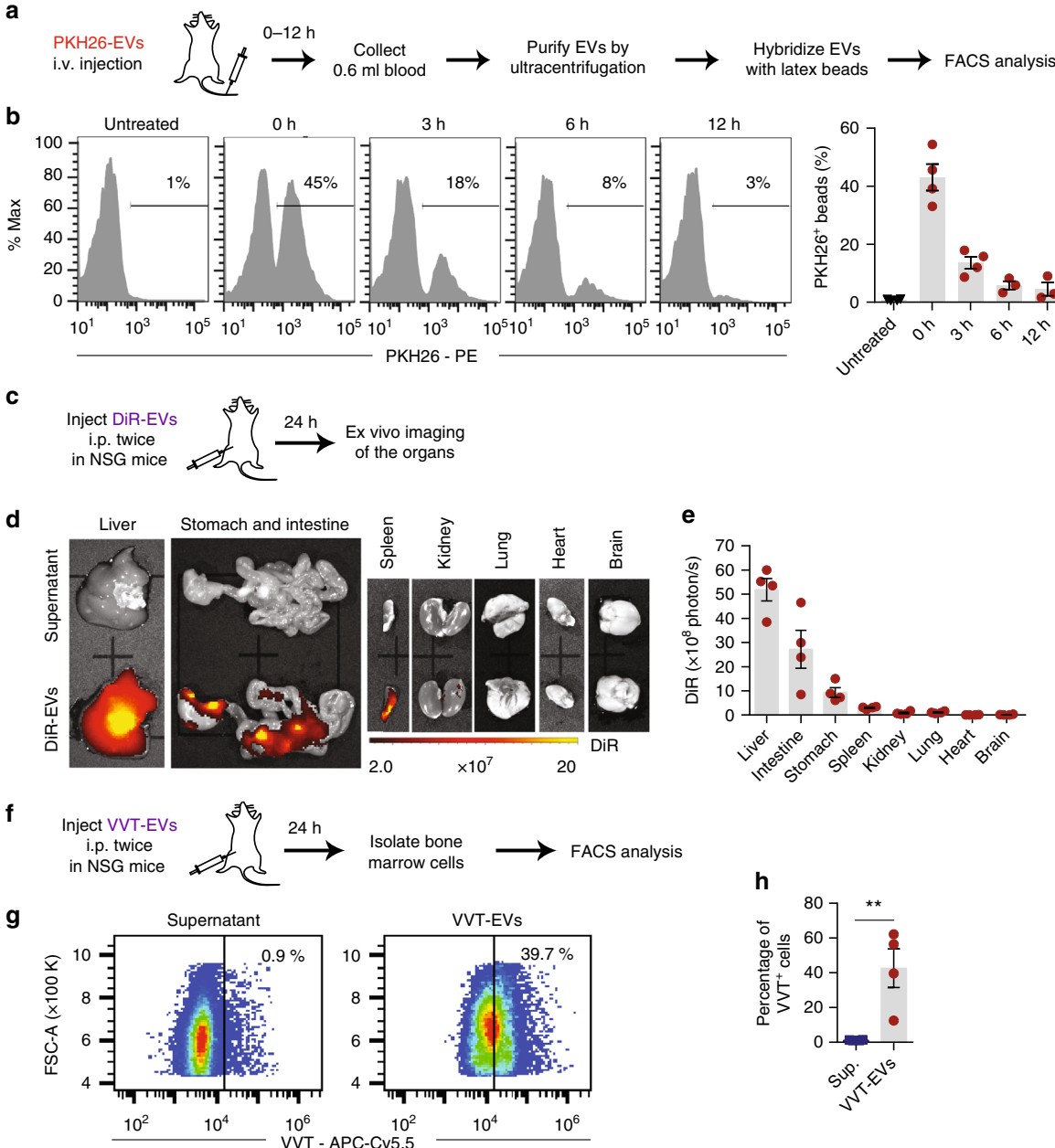

**Fig. 6** Biodistribution of RBCEVs upon systemic administration in NSG mice. **a** Experimental schema for determination of RBCEV circulation time following an i.v. injection. **b** FACS analysis of PKH26 fluorescence on the beads that were bound to total EVs from the blood of NSG mice immediately (0 h) or 3, 6, 12 h after the i.v. injection of $3.3 \times 10^{12}$ PKH26-labeled RBCEVs. The percentage of PKH26-positive beads are shown above the gate and the average is shown in the bar graph (mean ± SEM; $n = 3$ or 4 mice in two repeats). **c** Experimental schema for determination of RBCEV biodistribution in NSG mice. **d** Representative images of the organs 24 h after 2 i.p. injections (24 h apart) of $3.3 \times 10^{12}$ DiR-labeled RBCEVs or the supernatant from the last wash of labeled EVs. Images were captured using IVIS. DiR fluorescence is shown in pseudocolored radiance. **e** Average DiR radiance in the organs of the mice injected with DiR-labeled RBCEVs (mean ± SEM; $n = 4$ mice in 2 repeats). **f** Experimental schema for determination of vivotrack-680 (VVT)-labeled RBCEV distribution to the bone marrow in NSG mice. **g** FACS analysis of VVT fluorescence (APC-Cy5.5) vs. FSC-A of bone marrow cells from the mice 24 h after 2 i.p. injections (24 h apart) of $3.3 \times 10^{12}$ VVT-labeled RBCEVs or the EV wash supernatant (Sup). **h** Average percentage of VVT-positive cells (mean ± SEM, $n = 4$ mice in 2 repeats). $**P < 0.01$, one-tail Mann–Whitney test

## Discussion

Taken together, our data demonstrated the use of RBCEVs as a versatile delivery system for therapeutic RNAs, including short RNAs such as ASOs and gRNAs, as well as long RNAs such as *Cas9* mRNAs. ASOs and CRISPR–Cas9 can be designed and programmed to target any gene of interest, including undruggable targets such as oncomiRs and transcription factors, for therapeutic purposes. Previously, several research groups have

illustrated the advantages of using EVs for RNA delivery, but their EVs were generated from fibroblasts and dendritic cells that are not as readily available from all subjects[11,12]. EVs from whole plasma are more abundant and easier to obtain, but these EVs are derived from many cell types including nucleated cells, which still pose a risk for horizontal gene transfer[13].

Our RBCEV platform has several features that are more suitable for clinical applications. First, blood units are readily

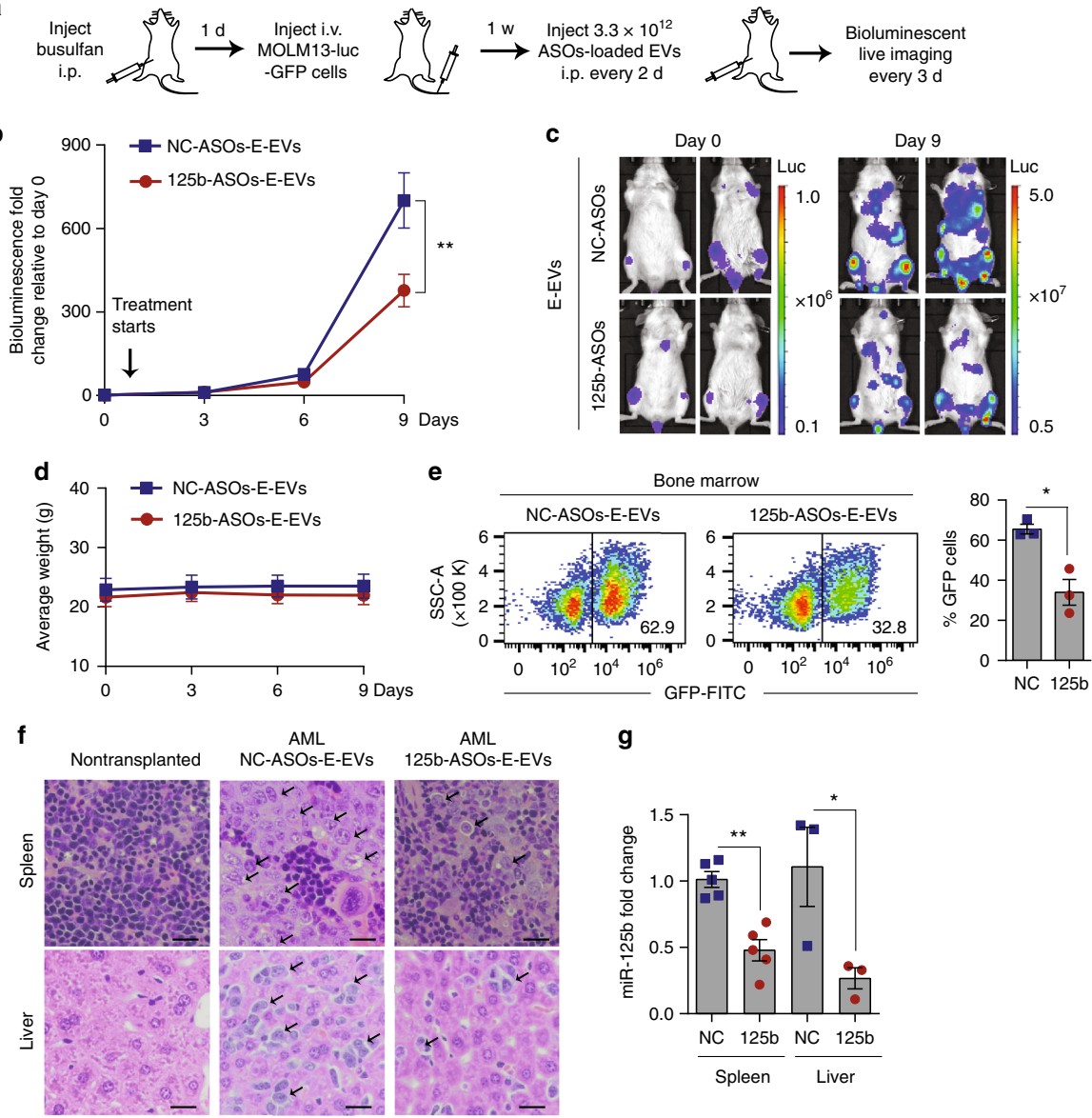

**Fig. 7** Systemic delivery of miR-125b ASOs in RBCEVs suppresses leukemia progression in AML xenografted mice. **a** Experimental schema of AML xenografting and ASOs delivery in NSG mice. **b** Average fold change in total body bioluminescence of the mice after 0 to 9 days of treatment with $3.3 \times 10^{12}$ RBCEVs containing NC-ASOs ($n = 7$ mice) or 125b-ASOs ($n = 6$ mice) relative to the signals before the treatment started (day 0), determined using an IVIS (mean ± SEM). **c** Representative images of the leukemic mice on day 0 & 9, captured using the IVIS. Bioluminescence is shown in pseudocolors. **d** Average weight of the mice (mean ± SEM). **e** FACS analysis of GFP cells in the bone marrow of the leukemic mice: representative dot plot of GFP (FITC channel) vs. size scatter area (SSC-A) and the average percentage of GFP-positive cells (mean ± SEM, $n = 3$ mice/group). **f** Representative H & E staining images of the spleen and liver from a nontransplanted mouse and from AML mice treated with NC/125b-ASOs-loaded RBCEVs. Arrows indicate clusters of infiltrating leukemia cells that have larger nuclei than normal cells. Scale bar, 50 µm. **g** miR-125b expression fold change normalized to U6B RNA in the spleen ($n = 5$ mice) and liver ($n = 3$ mice), determined using Taqman qRT-PCR and presented as mean fold change ± SEM relative to NC in the spleen. *$P < 0.05$; **$P < 0.01$ determined using one-tail Mann–Whitney test (**b**, **e**, **g**). The whole experiment was performed in two independent repeats

available from existing blood banks and even from the patients' own blood for allogeneic and autologous transfusion, respectively. A large number of RBCs ($\sim 10^{12}$ cells/L) are available in each blood unit. Hence, there is no need to expand the cells in culture and risk any accrual of mutations in vitro, and no cGMP-qualified media or supplements are required. Second, large-scale amounts ($10^{13}$–$10^{14}$) of EVs can be purified from RBCs, after the treatment with calcium ionophore, thus providing a scalable strategy to obtain EVs. Third, RBCEVs are safe, as the enucleated RBCs are homogeneously devoid of DNA, unlike EVs from nucleated cell types which pose potential risks for horizontal gene

transfer and unlike plasma EVs that are heterogeneous with unpredictable contents. For allogeneic treatments of cancer, RBCEVs are safer than plasma EVs, since cancer cells and immune cells are known to release a large amount of cancer-promoting EVs into the circulation of cancer patients[33,34]. Moreover, RBCEVs are nontoxic, unlike the synthetic transfection reagents. RNAs are stable in RBCEVs and fully functional in recipient cells as shown by our in vitro and in vivo data for liquid and solid cancers. RBCEVs are likely to be non-immunogenic, via matching of the blood types, unlike lentiviruses, adenoviruses, adeno-associated viruses, nanoparticles, and most synthetic

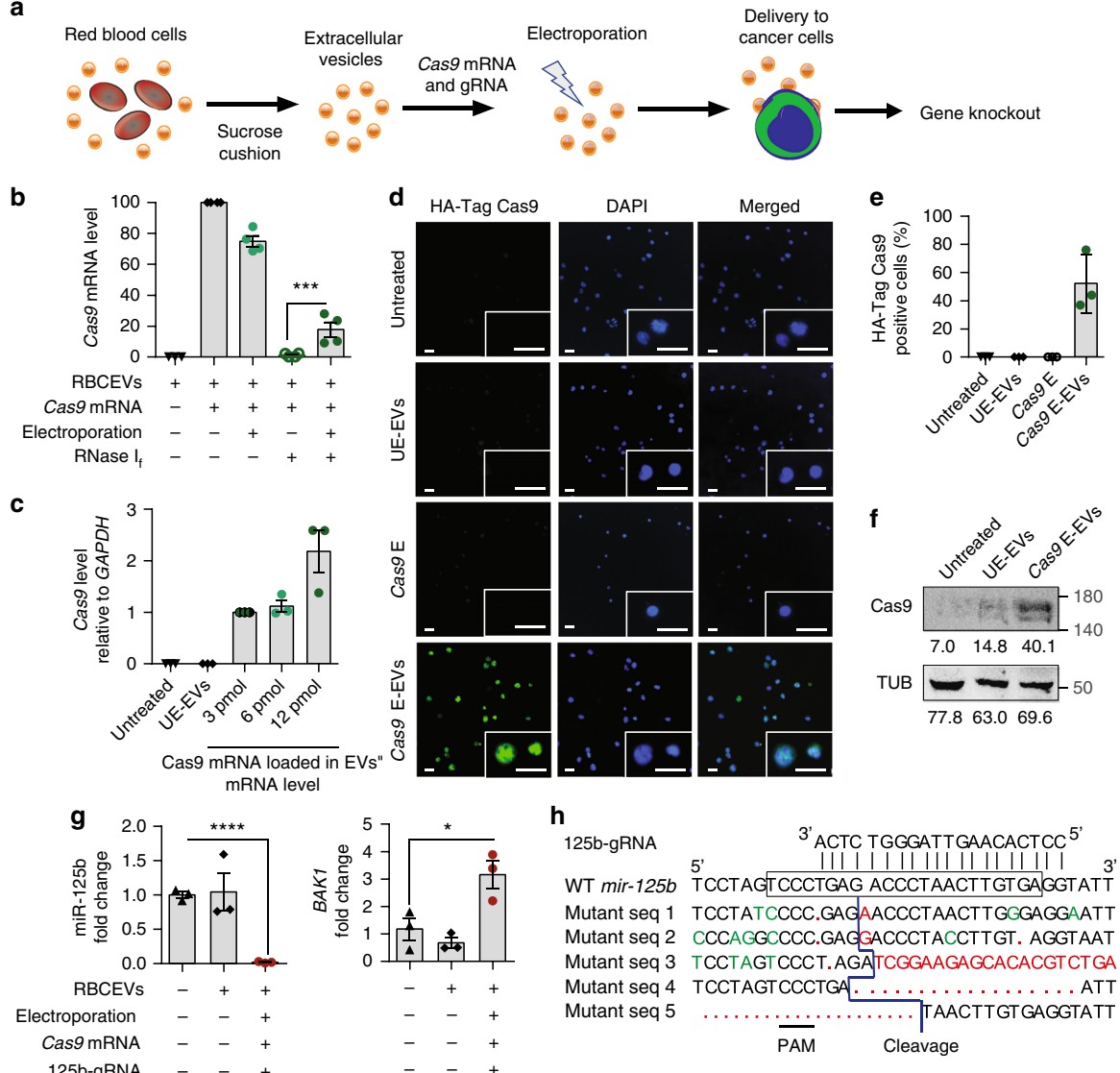

**Fig. 8** RBCEVs deliver *Cas9* mRNA and gRNA to leukemia cells for genome editing. **a** Schema of *Cas9* mRNA and gRNA delivery. **b** Average level of *Cas9* mRNA in $6.2 \times 10^{11}$ RBCEVs untreated, incubated or electroporated with 6 pmol *Cas9* mRNA and treated with RNase $I_f$, relative to the unelectroporated *Cas9* level (2nd condition). **c** The level of *Cas9* mRNA relative to *GAPDH* mRNA in MOLM13 cells that were incubated with $12.4 \times 10^{11}$ unelectroporated RBCEVs (UE-EVs) or RBCEVs electroporated with 3, 6, or 12 pmol *Cas9* mRNA (E-EVs) after 24 h of treatment, relative to the 3 pmol condition. **d** Representative images of MOLM13 cells that were incubated for 48 h with $12.4 \times 10^{11}$ UE-EVs or EVs that were electroporated with 6 pmol *Cas9* mRNAs. MOLM13 cells were also electroporated directly with 6 pmol *Cas9* mRNAs (*Cas9* E) for comparison. The cells were stained for HA-Cas9 protein (green) and nuclear DNA (Hoechst, blue). Scale bar, 20 μm. **e** Average percentage of MOLM13 cells stained positive for HA-Cas9 protein as shown in **d**. **f** Western blot analysis of Cas9 and α-tubulin (TUB) in MOLM13 cells untreated, treated with $12.4 \times 10^{11}$ unelectroporated or 6 pmol-*Cas9* mRNA-loaded RBCEVs. Below each band is its mean intensity, quantified using ImageJ. **g** miR-125b and *BAK1* expression fold change, relative to untreated condition, normalized to *U6b* RNA and *18s* RNA respectively, in MOLM13 cells treated with $12.4 \times 10^{11}$ UE-EVs or EVs loaded with 6 pmol *Cas9* mRNA and *mir-125b*-targeting gRNA for 48 h. **h** Alignment of *mir-125b*-targeting gRNA with wildtype (WT) *mir-125b* (frame indicates mature sequence) and mutant DNA sequences from MOLM13 cells treated as in **g**. Red, insertion or deletion. Green, mismatch. PAM protospacer adjacent motif. All the bar graphs are presented as mean ± SEM ($n = 3$ or 4 repeats of 3 or 4 cell passages). *$P < 0.05$; ***$P < 0.001$; ****$P < 0.0001$: one-tail Student's *t*-test

transfection reagents. And we have also shown that RBCEVs deliver RNAs to cells at a higher efficiency than two commonly used transfection reagents. We are able to deliver not only short RNAs but also long mRNAs in RBCEVs. RBCEVs were successfully used to target a specific oncomiR gene, not only via steric blocking, but also via CRISPR–Cas9 genome editing, which has not been shown hitherto. RBCEVs from group O− Rh negative blood are ideal for universal treatments. Finally, the fact that RBCEVs can be frozen and thawed many cycles without affecting their integrity and efficacy, suggests that they can be developed into stable pharmaceutical products in future. Further development of RBCEVs coated with cancer-targeting peptides or antibodies could potentially deliver therapeutic RNAs to cancer cells specifically, and reduce adverse side effects in normal tissues.

## Methods

**Cell culture**. Acute myeloid leukemia MOLM13 and NOMO1 cells were originally from DSMZ Collection of Microorganisms and Cell Cultures (Braunschweig, Germany). Breast cancer MCF10CA1a (CA1a) cells were purchased from Karmanos Cancer Institute (Wayne State University, USA). HEK-293T cells were

obtained from the American Type Culture Collection (ATCC, USA). MCF10CA1a cells with luciferase label were generated in our previous study[35]. 293T-eGFP cells, generated from ATCC HEK-293T cells, were a gift from Dr. Albert Cheng (Jackson Lab, USA). MOLM13 and NOMO1 cells stably expressing eGFP were generated by an infection with pCAG-eGFP lentivirus generated in HEK-293T cells cotransfected with packaging plasmids using Lipofectamin$^{TM}$ 2000 (plasmids from Addgene, USA) and sorted using flow cytometry. Leukemia and breast cancer cell lines were maintained in RPMI1640 or DMEM (ThermoFisher Scientific) respectively with 10% fetal bovine serum (Biosera, USA) and 1% penicillin/streptomycin (ThermoFisher Scientific, USA). All the cells used for the experiments were tested negative for mycoplasma contamination using PCR. Briefly, 100 μl supernatants were collected from 80–100% confluent cultures and heated at 95 °C for 5 min. Mycoplasma DNA was amplified using *Taq* polymerase (a gift from the Xia lab, Xiamen University) and a mycoplasma-specific primer mixture (sequences provided below) in 35 cycles: 94 °C 15 s, 56 °C 15 s, 72 °C 30 s; and visualized using 1% agarose gel. In parallel, mycoplasma was also tested using the Mycofluor mycoplasma detection kit (ThermoFisher Scientific) according to the manufacturer's protocol. MOLM13, NOMO1, CA1a, and 293T cells were authenticated by their original vendors. We also confirmed the identity of these cell lines and their fluorescent/luciferase derivatives using a 16-loci multiplex short tandem repeat analysis according to ATCC standards, provided by an authentication service by Guangzhou IGE Biotechnology (China).

**EV purification from RBCs.** Group O blood samples were obtained by Red Cross from healthy donors in Hong Kong with informed consents. All experiments with human blood samples were performed according to the guidelines and the approval of the City University of Hong Kong Human Subjects Ethics committee. RBCs were separated from plasma and white blood cells by using centrifugation and leukodepletion filters (Terumo Japan). Isolated RBCs were diluted in PBS and treated with 10 mM calcium ionophore (Sigma Aldrich) overnight. To purify EVs, RBCs and cell debris were removed by centrifugation at $600 \times g$ for 20 min, $1600 \times g$ for 15 min, $3260 \times g$ for 15 min, and $10,000 \times g$ for 30 min at 4 °C. The supernatants were passed through 0.45 μm-syringe filters. EVs were concentrated by using ultracentrifugation with a TY70Ti rotor (Beckman Coulter, USA) at $100,000 \times g$ for 70 min at 4 °C. EVs were resuspended in cold PBS. For labeling, half of the EVs were mixed with 20 μM PKH26 (Sigma Aldrich, USA) or 1 μM DiR (ThermoFisher Scientific) or 42.62 μM Vivo-Track-680 (Perkin Elmer). Labeled or unlabeled EVs were layered above 2 ml frozen 60% sucrose cushion and centrifuged at $100,000 \times g$ for 16 h at 4 °C using a SW41Ti rotor (Beckman Coulter) with reduced braking speed. The red layer of EVs (above the sucrose) was collected and washed once (unlabeled EVs) or twice (labeled EVs) with cold PBS using ultracentrifugation in a SW41Ti rotor (Beckman Coulter) at $100,000 \times g$ for 70 min at 4 °C. All ultracentrifugation experiments were performed with a Beckman XE-90 ultracentrifuge (Beckman Coulter). Purified RBCEVs were stored at −80 °C.

**EV characterization.** The concentration and size distribution of EVs were quantified using a NanoSight Tracking Analysis NS300 system (Malvern, UK). Zeta potential and polydispersity index were determined using a Zetasizer Nano (Malvern). For transmission electron microscopy analysis of EVs, EVs were fixed on copper grids (200 mesh, coated with formvar carbon film) by adding equal amount of 4% paraformaldehyde. After washing with PBS, 4% uranyl acetate was added for chemical staining of EVs and images were captured using a Tecnai 12 BioTWIN transmission electron microscope (FEI/ Philips, USA).

**Oligonucleotide sequences and modifications.** Anti-miR-125b ASOs (5′-UCACAAGUUAGGGUCUCAGGGA-3′) and negative control ASOs (5′-CAGUACUUUUGUGUAGUACAA-3′) were synthesized with 2′ O-methyl modification at every ribonucleotide by Shanghai Genepharma (Shanghai, China) or by Integrated DNA Technology (Singapore). Anti-miR-125b gRNA (5′-CCUCACAAGUUAGGGUCUCA- Synthego Scaffold −3′) and anti-GFP gRNA: (5′-GGGCACGGGCAGCUUGCCGG- Synthego Scaffold −3′) were synthesized with 2′-O-methyl analogs and 3′ phosphorothioate internucleotide linkages at the first three 5′ and 3′ terminal RNA residues by Synthego (USA).

**EV electroporation.** Electroporation of RBCEVs were performed using a Gene Pulser Xcell electroporator (BioRad), exponential program at a fixed capacitance of 100 μF with 0.4 cm cuvettes. For optimization, $8.25$ to $16.5 \times 10^{11}$ RBCEVs were diluted in OptiMEM (ThermoFisher Scientific) and mixed with 4 μg Dextran conjugated with Alexa Fluor® 647 (AF647, ThermoFisher Scientific) to a total volume of 200 μl. An aliquot of 100 μl EV mixture was added to each cuvette and incubate on ice for 10 min. Electroporation was tested at different voltages: 50–250 V. For ASOs delivery, $\sim 4–16 \times 10^{11}$ RBCEVs were electroporated with 400 pmol scrambled negative control (NC) or anti-miR-125b ASOs at 250 V. For genome editing, $12.4 \times 10^{11}$ RBCEVs (or MOLM13 cells) were electroporated with 6 pmol CleanCap$^{TM}$ *Cas9* mRNA (Trilink) and 50 pmol anti-GFP gRNA or 80 pmol anti-*mir-125b* gRNA at 400 V. Aggregates of RBCEVs formed during electroporation were dissolved by vigorous pipetting. To quantify the electroporation efficiency, $8.25 \times 10^{11}$ Dextran-AF647 electroporated EVs were incubated overnight with 5 μg

latex beads (ThermoFisher Scientific) and analyzed for AF647 using flow cytometry.

**RNA loading efficiency and stability in electroporated RBCEVs.** To quantify the amount of unbound ASOs, 200 pmol unlabeled or FAM-labeled NC-ASOs were loaded with or without $8.25 \times 10^{11}$ RBCEVs, with or without electroporation, into 10% Tris-acetate-EDTA (TBE) native gel, separated at 150 V for 30 min and visualized using SYBR-Gold staining for 30 min at room temperature (Thermo-Fisher Scientific) or using FAM fluorescence, respectively with the Gel Doc$^{TM}$ EZ Documentation system (Bio-Rad, USA). The experiment was repeated three times independently. The SYBR Gold bands of the ASOs were quantified using imageJ (NIH, USA) and normalized to the background. Full images of the gels are provided in Supplementary Fig. 17.

To determine the stability of ASOs in electroporated EVs, $6.2 \times 10^{11}$ of RBCEVs and 200 pmol FAM ASOs unelectroporated or electroporated mixtures were incubated with 50% FBS in OptiMEM at 37 °C for 1–72 h. The mixtures were added into a 96-well black plate with clear bottom (Perkin Elmer, USA) and FAM fluorescence was analyzed using a Synergy$^{TM}$ H1 microplate reader (BioTek, USA).

To quantify the efficiency of ASOs electroporation, $6.2 \times 10^{11}$ of RBCEVs electroporated with 200 pmol of 125b-ASOs or the same amount of unelectroporated EVs and 125b-ASOs were incubated with 100 units of RNase I$_f$ (New England Biolabs, USA) at 37 °C for 30 min. The RNase was heat inactivated by incubating at 70 °C for 10 min. Trizol-LS (ThermoFisher Scientific) was added into each sample and the extracted RNA was reverse transcribed as described below. Similarly, $6.2 \times 10^{11}$ RBCEVs electroporated with 1 μg of *Cas9* mRNA or the same amount of unelectroporated *Cas9* mRNA were incubated with 25 units of RNaseI$_f$ at 37 °C for 5 min and subjected to RNA extraction and qRT-PCR of *Cas9* mRNA.

**EV separation using top-down sucrose density gradients.** A total of $3.7 \times 10^{12}$ FAM ASOs-loaded EVs were mixed with 1 ml of 10% HEPES/sucrose solution. An 11 ml of linear sucrose gradient (60–10%) was loaded into a 12.5 ml-open top SW41 ultracentrifuge tube (Beckman Coulter). The EV suspension was layered on top of the sucrose layer and ultracentrifuged at $150,000 \times g$ for 16 h at 4 °C. A total of 12 fractions were collected from the sucrose gradient into a black well plate and analyzed using the Synergy™ H1 Microplate Reader. The concentration of EVs in each fraction was determined using the NanoSight as described above. The density of sucrose in each fraction was measured using a refractometer (VastOcean, China).

**Treatment of cancer cells with RBCEVs in vitro.** We performed quality control of RBCEVs for every batch of purification using a Nanosight particle analyzer. Samples with unusually low concentration or strange aggregates were discarded. Cells in culture were routinely examined for signs of contamination or changes in morphology and growth. To test the EV uptake, 50,000 MOLM13, CA1a or NOMO1 cells were incubated with 200 μl of $\sim 4–16 \times 10^{11}$ unelectroporated or electroporated EVs and 300 μl growth medium per well in 24-well plates for 24 h. Untreated controls were kept in the same medium with 200 μl untreated Opti-MEM. For assays that required longer incubation time, the medium was replaced with fresh growth medium after 24 h. For comparison, MOLM13 cells were transfected with 4 μg Dextran-AF647 or 800 nM FAM-labeled NC-ASOs in Lipofectamin$^{TM}$ 3000 (ThermoFisher Scientific) or INTERFERin® (PolyPlus Transfection, France) according to the manufacturers' protocols. To test the effect of heparin on the uptake, MOLM13 cells were pretreated with 20 μg/ml of heparin sodium salt (Aladdin, China) for 10 min then incubated overnight with $12.4 \times 10^{11}$ of unlabeled or PKH26-labeled RBCEVs in the presence or absence of 20 μg/ml of heparin sodium salt. The uptake of RBCEVs and Dextran or FAM ASOs were analyzed using flow cytometry or immunostaining.

**Flow cytometry analysis.** Flow cytometry of latex beads or cells in FACS buffer (PBS containing 0.5% fetal bovine serum) was performed using a CytoFLEX-S cytometer (Beckman Coulter) or SH800Z cytometer (Sony Biotechnology, USA) and analyzed using Flowjo V7 or V10 (Flowjo, USA). GFP-positive MOLM13 or NOMO1 cells were selected using a Sony SH800Z cell sorter in sterile condition. The beads or cells were initially gated based on FSC-A and SSC-A to exclude the debris and dead cells (low FSC-A) as shown in Supplementary Figs. 3, 16. The cells were further gated based on FSC-width vs. FSC-height, to exclude doublets and aggregates. In the analysis of GFP + cells from the bone marrow of leukemic NSG mice, the live cells were also gated from the single cells population based on Cytox blue negative (PB450 channel). Subsequently, the fluorescent-positive beads or cells were gated in the appropriate fluorescent channels: FITC for FAM and GFP, PE for PKH26 and PI, APC for AF647, and APC-Cy5.5 for VVT, as the populations that exhibited negligible signals in the unstained/untreated negative controls as shown by the example in Supplementary Figs. 3, 16. FCS and analysis files are provided in figshare.

**Western blot analysis.** Total cell lysates were extracted from EVs or cells (cells from 3 wells of 24-well plates were combined for each condition) by incubating with RIPA buffer supplemented with protease inhibitors (Biotool). A total of 30 or

35 μg of protein lysates were separated on 10% polyacrylamide gels and transferred to a Nitrocellulose membrane (GE Healthcare). PM5100 ExcelBand™ 3-color high range protein ladder (SmoBio, Taiwan) was loaded at two sides of the samples. Membranes were cut horizontally into two to four pieces based on the ladder, blocked with 5% non-fat milk in Tris buffered saline containing 0.1% Tween-20 (TBST) for 1 h at room temperature and incubated with primary antibodies overnight at 4 °C: mouse anti-Alix (clone 3A9, Cat # SC-53538, Santa Cruz, dilution 1:500), mouse anti-Tsg101 (clone C-2, Cat # SC-7964, Santa Cruz, dilution 1:500), rabbit anti-Hemoglobin α (clone H-80, SC-21005, Santa Cruz, dilution 1:1000), mouse anti-Calnexin (clone AF18, SC-23954, Santa Cruz, dilution 1:500), mouse anti-Stomatin (clone E-5, SC-376869, Santa Cruz, dilution 1:1000), rabbit anti-GAPDH (clone FL-335, Cat # SC-25778, Santa Cruz, dilution 1:1000), mouse anti-Cas9 antibody (clone 7A9, Cat # 844301, BioLegend, dilution 1:250), and mouse anti-tubulin (clone DM1A, Cat # Ab7291, Abcam, dilution 1:1000). The blot was washed three times with TBST then incubated then with HRP-conjugated anti-mouse and anti-rabbit secondary antibodies (Cat # 715-035-150 and 711-035-152, Jackson ImmunoResearch, dilution 1:10,000,) for 1 h at room temperature. The blot was imaged using an Azure Biosystems gel documentation system. Full images of the blots are provided in Supplementary Fig. 18. The intensity of the bands were quantified and subtracted by the background using ImageJ.

**RNA extraction and qRT-PCR.** Total RNA was extracted from cells or tissues using Trizol (ThermoFisher) according to the manufacturer's manuals. RNA samples were quantified and qualified using NanoDrop analysis (ThermoFisher) and 1% agorose gels. Those with sufficient concentration (>30 ng/μl), purity (A260/280 > 1.7) and integrity (clear 28 and 18s rRNA bands) were converted to cDNA using a high capacity cDNA reverse transcription kit (ThermoFisher) following the manufacturer's protocol. The levels of miR-125a and miR-125b were quantified using Taqman® miRNA assays (ThermoFisher), normalized to the expression of U6b snRNA, or using the miRCURY™ LNA™ Universal RT microRNA PCR (Qiagen, Germany), normalized to the expression of miR-103a. Anti-miR-125b ASOs was quantified using a Taqman® miRNA assay (ID #007655_mat), normalized to the expression of U6b snRNA or its copy number was calculated based on a standard curve of ASOs threshold cycle (Ct) values vs. the concentration of the ASOs in a serial dilution from 0.001 to 1000 pM. qRT-PCR analysis of mRNAs was performed using Ssofast™ Evagreen (SYBR Green) qPCR master mix (Bio-Rad), normalized to the expression of *GAPDH*, or *18s rRNA*, or *ACTB*. All qPCR reactions were performed by using a CFX96 Touch™ Real-Time PCR Detection System (Bio-Rad).

**Sequencing the *mir-125b* gene.** To identify the genome editing events generated by *mir-125b* targeting gRNA and *Cas9* mRNA-loaded RBCEVs, DNA were extracted from the EV treated MOLM13 cells using Trizol followed by phenol-chloroform and ethanol precipitation. The primary *hsa-mir-125b-2* sequence (383 nucleotides) was amplified using a pair of gene-specific primers (sequences provided below) and Q5® Hot Start high-fidelity master mix (New England Biolabs) in 30 cycles: 98 °C 10 s, 64 °C 30 s, 72 °C 45 s, and a final extension of 72 °C for 2 min. The PCR product was reamplified using the same primers with sequencing tags. High-through sequencing was performed by NovoGene Sequencing company (Hong Kong) using the HiSeq paired-end platform (Illumina, USA).

**Primer sequences.** *GAPDH* Forward: GGAGCGAGATCCCTCCAAAAT
*GAPDH* Reverse: GGCTGTTGTCATACTTCTCATGG
*18s RNA* Forward: GTAACCCGTTGAACCCCATT
*18s RNA* Reverse: CCATCCAATCGGTAGTAGCG
*BAK1* Forward: TGGTCACCTTACCTCTGCAA
*BAK1* Reverse: TCATAGCGTCGGTTGATGTC
*SP-Cas9* Forward: AAGGGACAGAAGAACAGCCG
*SP-Cas9* Reverse: ATATCCCGCCCATTCTGCAG
*mir-125b* Forward: AATGGTCGTCGTGATTACTCA
*mir-125b* Reverse: TTTTGGGGATGGGTCATGGT
*ACTB* Forward: TCCCTGGAGAAGAGCTACGA
*ACTB* Reverse: AGGAAGGAAGGGTGGAAGAG
Mycoplasma-1 Forward: CGCCTGAGTAGTACGTTCGC
Mycoplasma-2 Forward: CGCCTGAGTAGTACGTACGC
Mycoplasma-3 Forward: TGCCTGAGTAGTACATTCGC
Mycoplasma-4 Forward: TGCCTGGGTAGTACATTGAC
Mycoplasma-5 Forward: CGCCTGGGTAGTACATTGAC
Mycoplasma-6 Forward: CGCCTGAGTAGTATGCTCGC
Mycoplasma-1 Reverse: GCGGTGTGTACAAGACCCGA
Mycoplasma-2 Reverse: GCGGTGTGTACAAAACCCGA
Mycoplasma-3 Reverse: GCGGTGTGTACAAACCCCGA
All primers were synthesized by Beijing Genomics Institute (China).

**Quantification of cell viability.** For flow cytometry analysis, leukemia cells were stained with Propidium iodide (PI) for 15 min at room temperature, washed with FACS buffer and analyzed using a CytoFLEX-S cytometer as described above. For a plate assay, 50,000 CA1a cells were seeded per well in 24-well plates and treated with ASO-electroporated RBCEVs. After 3 days, the cells were washed once with PBS, and incubated with 0.5% crystal violet staining solution (Sigma Aldrich). Plates were then washed three times in a stream of tap water and air dried. Afterwards, 500 μl of 50% acetic acid was added into each well and the optical density was measured at 570 nm using the Biotek micro-plate reader.

**Animal experiments.** All mouse experiments were performed according to our experimental protocols approved by the Animal Ethics Committees at City University of Hong Kong and the Department of Health, Hong Kong SAR, complied with the government legislations including the Animals (Control of Experiments) Ordinance (Cap. 340) and the Prevention of Cruelty to Animals Ordinance (Cap. 169). Nude mice (strain NU/J 002019) and NSG mice (NOD.CgPrkdc < scid > Il2rg < tm1Wjl > /SzJ 005557) were purchased from the Jackson Laboratory (USA) and bred in our facilities. Mice of similar ages were labeled with numbers on their ear tags or tails using permanent markers and randomized into control and treatment groups without any bias on parents, weight, size, or gender (except for the breast cancer experiments that were performed in female mice only). Most of the imaging and histopathology experiments were done in a blinded fashion as the researchers who performed the data collection were different from those who performed the treatments. The data were recorded based on the mouse numbers, not by their treatment groups. Mice that became pregnant or died accidentally due to anesthesia during the experiments were excluded. We also excluded false positives or negatives due to other technical issues such as unsuccessful injections.

**Determination of EV uptake and biodistribution in vivo.** A total of $5 \times 10^6$ CA1a cells in 50 μl PBS were mixed with an equal volume of cold Matrigel (Corning) and injected subcutaneously in the left and right flanks of 6-week-old female nude mice. After 1 week, when the tumors approach 7 mm in diameter, each mouse was injected with $16.5 \times 10^{11}$ PKH26-labeled RBCEVs intratumorally in the left flank. The mice were subsequently fed with alfalfa-free diet (LabDiet, USA) to reduce autofluorescence in the digestive system. PKH26 fluorescence was measured every day using the IVIS Lumina III system (Caliper Life Sciences, USA). On day 3, mice were killed and the tumors were excised and imaged for PKH26. Frozen sections of the tumors were stained with DAPI and observed under a LSM-880 NLO confocal laser scanning microscope (Zeiss, Germany). Similarly, the CA1a tumor (7 mm) bearing mice were injected i.p. with $16.5 \times 10^{11}$ PKH26 or DiR-labeled RBCEVs or the supernatant of the last wash after EV labeling. Images of DiR-EV injected mice were captured at 24 h post-treatment using the IVIS Lumina III. Tumors, livers, hearts, lungs, spleens, kidneys, stomach, and intestines were also obtained from i.p. DiR/PKH26-EV-injected mice for IVIS imaging and histopathology analysis.

To determine the biodistribution of RBCEVs in NSG mice, $3.3 \times 10^{12}$ DiR-labeled RBCEVs or the supernatant of the last EV wash were injected twice with 24 h interval in 6–8-week-old NSG mice (Jackson Lab). Thereafter, the mice were killed and the organs were collected for DiR fluorescence imaging using the IVIS Lumina III system. Whole body and tissue fluorescence images were acquired and analyzed using Living Image® software (Caliper Life Sciences). The background and autofluorescence were defined using the untreated or supernatant negative controls and subtracted from the images using Image-Math function.

To determine the uptake of RBCEVs by cells in the bone marrow, $3.3 \times 10^{12}$ VVT-labeled RBCEVs or the supernatant of the last EV wash were injected twice with 24 h intervals in 6–8-week-old NSG mice (4 mice/group). 24 h after the second injection, the mice were killed and their bone marrow was collected by flushing FACS buffer through the crushed bones with a syringe and G25 needle. The cells were filtered through a 70 μm strainer, centrifuged at 1500 rpm for 5 min, resuspended in FACS buffer and analyzed for VVT fluorescence (APC-Cy5.5 channel) using the Sony SH800Z cytometer as described above.

**Stability of RBCEVs in the circulation.** A total of $3.3 \times 10^{12}$ PKH26-labeled EVs were injected i.v. into the tail veins of 6-week NSG mice (Jackson Lab). The mice were killed immediately or 3, 6, and 12 h after the injection. The blood was collected from the heart into the EDTA tubes and centrifuged at $800 \times g$ for 5 min. The supernatant was diluted with 10 ml cold PBS and centrifuged at $10,000 \times g$ for 15 min and passed through a 0.45 μm filter to remove the debris. Afterwards, the EVs were ultracentrifuged at $100,000 \times g$ for 90 min at 4 °C. Purified EVs were resuspended in 175 μl PBS and incubated with 2.5 μg of latex beads at 37 °C for 30 min, followed by an overnight incubation at 4 °C. EV-bound latex beads were washed with 1 ml FACS buffer at $4000 \times g$ for 10 min and PKH26 fluorescence was analyzed using the CytoFLEX-S cytometer (Beckman).

**In vivo delivery of ASOs to breast tumors.** A total of $5 \times 10^5$ luciferase-labeled CA1a cells were mixed with equal volume of matrigel and injected into the flanks of 6-week-old female nude mice. After 24 h, each mouse was injected with $8.25 \times 10^{12}$ NC/125b-ASO-loaded (n = 8 mice) or 400 pmol unbound NC/125b-ASOs (n = 6 mice) subcutaneously in the flank at the same site where the tumor cells were injected. Intratumoral injections of the ASO-electroporated EVs were repeated every 3 days until day 42. Bioluminescent images of the whole body were taken every 6 days using the IVIS Lumina III system following i.p. injection of 150 mg/kg D-luciferin (Caliper Life Sciences). All bioluminescent images were acquired and analyzed using Living Image® 4.5. software in a blinded manner (Caliper Life Sciences). On day 44, the mice were killed and the tumors were excised. Half of

each tumor was homogenized in Trizol for RNA extraction and qRT-PCR of miR-125b, except the RNA samples from a few tumors that did not meet the quality controls (as described above). The other half of the tumor and the lung was fixed for histopathology analysis as described below.

**In vivo delivery of ASOs to leukemia engrafted mice**. NSG mice of 7–8-week-old were injected i.p. with 20 mg/kg Busulfan (Santa Cruz). After 24 h, $5 \times 10^5$ luciferase and GFP labeled MOLM13 cells were injected into the tail vein of the busulfan-conditioned mice. A week later, bioluminescence was measured using the IVIS Lumina III. Mice with successful engraftment of leukemia cells (shown by bioluminescent signals in the bone marrow) were treated with $3.3 \times 10^{12}$ ASO-loaded EVs i.p. every other day for 9 days (5 doses in total). Luminescence images of whole body were taken every 3 days using the IVIS Lumina III system following i.p. injection of 150 mg/kg D-luciferin in a blinded manner (Caliper Life Sciences). After 9 days of treatment, the mice were killed. Bone marrow were collected and analyzed for GFP-positive cells using FACS. Briefly the cells were collected from the femur or tibiae of killed mice by flushing FACS buffer through the crushed bones with a syringe and G25 needle. The cells were filtered through a 70 μm strainer, centrifuged at 1500 rpm for 5 min, resuspended in 0.5 ml RBC lysate buffer (0.8% NH4Cl and 0.1 mM EDTA in water buffered with KHCO3 to achieve a final pH of 7.2–7.6) and incubated on ice for 5 min. The buffer was neutralized with 5 ml DMEM containing 10% FBS. The cells were centrifuged again, washed once with PBS and resuspended in 200 μl FACS buffer. 0.5 μl Cytox blue (ThermoFisher Scientific) was added for identification of dead cells. The spleen and liver were collected in Trizol or formalin for RNA extraction and histopathology analysis, respectively. The RNA samples were used for qRT-PCR of miR-125b, except those that did not meet the quality controls (as described above).

**Histopathology**. Tumors and other tissues from the mice were fixed in 10% buffered formalin (ThermoFisher Scientific) overnight at room temperature, dehydrated sequentially in 70, 95, and 100% alcohol at 37 °C, cleared in three baths of xylene (ThermoFisher Scientific) and impregnated in three baths of paraffin wax (ThermoFisher Scientific) each for 1 h and 30 min at 37 and 62 °C, respectively. Paraffin blocks were cut at 5 μm using a microtome (MICROM model: HM330). Sections were dried in a 37 °C incubator before staining. Sections were dewaxed in two baths of xylene, then immersed in two baths of absolute alcohol and one bath of 70% alcohol, each for 10 min. Sections were rehydrated in water and stained with Gill's haematoxylin (Surgipath) for 15 min. After washing with water, stained sections were differentiated in 0.3% acid alcohol, washed in water again and blued in 2% sodium bicarbonate solution. Microscopic examination is essential to check distinct nuclei staining with clean cytoplasm and background. Sections were then stained with 0.5% Eosin for 2 min. After a quick wash in water to remove excess Eosin (Merck), sections were dehydrated in 95% and absolute alcohol. Sections were then cleared in xylene and mounted with a synthetic mountant (Shandon).

**Immunostaining**. MOLM13 cells were fixed with 4% paraformaldehyde (Sigma Aldrich) and adhered to glass slides using cytospin at 400 rpm for 3 min. The cells were blocked with 5% normal donkey serum (Jackson Immuno Research), permeabilized with 0.2% Triton X-100, and incubated overnight with a mouse anti-HA antibody (Clone F-7, Cat #7392, Cell Signaling Technology, 1:250 dilution) at 4 °C, washed three times with PBS and incubated with Alexa Fluor® 488 donkey anti-mouse secondary antibody (Jackson Immuno Research) for 1 h at room temperature. The cells were finally stained with Hoechst (Sigma Aldrich) for 5 min at room temperature and washed three times with PBS. In the EV uptake experiment (Fig. 1f), the cells were stained with Hoechst only but not with any antibody. Images were captured using Nikon Eclipse Ni-E upright fluorescence microscope. Images were analyzed using ImageJ. The number of Cas9-HA positive nuclei were counted and normalized by the number of Hoechst positive nuclei in the same image. The average percentage of Cas9-HA positive cells was calculated from three samples in each treatment.

**Statistical analysis**. Student's t-tests, calculated using Microsoft Excel, were used to compute the significance between the treated samples and the controls; the test type was set to one-tail distribution and two-sample equal variance. The Mann–Whitney test (one-tail), computed using GraphPad Prism 6, was employed to calculate the significance where the data did not follow a normal distribution. One-way ANOVA, calculated using GraphPad Prism 6, was used for analysis of data involving multiple groups of treatments. All P-value < 0.05 were considered significant. In all graphs, data are presented as mean ± standard error of the mean (SEM). For quantification, each experiment was usually repeated thre times with RBCEVs from three donors or with cells from three cell passages. Mouse experiments were performed with groups of 3 to 8 mice. The minimum sample size of 3 was determined using G*Power analysis for one-tail t-test comparing the mean difference of two independent groups with effect size $d = 5$; α err prob = 0.05 and power = 0.95.

**Data availability**. Data generated in this study are available in figshare.com including raw data supporting Fig. 1 (https://doi.org/10.6084/m9.figshare.5931136), Fig. 2 (https://doi.org/10.6084/m9.figshare.5923570),

Fig. 3 (https://doi.org/10.6084/m9.figshare.593113), Fig. 4 (https://doi.org/10.6084/m9.figshare.5931724), Fig. 5 (https://doi.org/10.6084/m9.figshare.5931733), Fig. 6 (https://doi.org/10.6084/m9.figshare.5931970), Fig. 7 (https://doi.org/10.6084/m9.figshare.5932261) and Fig. 8 (https://doi.org/10.6084/m9.figshare.5932654). Next-generation sequencing data of mir-125b mutants are available in Bioproject database: ID # PRJNA433779. All other data are available from the authors upon reasonable request.

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

## Acknowledgements

We sincerely thank Hong Kong Red Cross for providing blood samples. Many thanks to Dr. Linfeng Huang, Dr. Zongli Zheng, Dr. Leo Chan, Dr. Liang Zhang and Dr. Gigi Lo (City University of Hong Kong), Prof. Harvey Lodish (Whitehead Institute), Dr. Pierre-Yves Mantel (University of Fribourg), Prof. Judy Lieberman (Harvard Medical School), Dr. Albert Cheng (Jackson Lab), Dr. Chinh Duong (Vietnam National Institute of Hematology and Blood Transfusion), Prof. Randy Schekman (University of California Berkley), Dr. Wai Leong Tam (Genome Institute of Singapore), Prof. Ningshao Xia and Prof. Shengxiang Ge (Xiamen University), and Prof. Andrew Grimson (Cornell University) for providing valuable reagents and advices. We also thank Lawrence Chan, Seongkyeol Kim, Theodoros Kiomourtzis, Ching Yee Moo, Tianzhong Li, Limin Feng, Thach Tuan Pham, Dr. Amy Fong, Xin Zhang, Abdullah Faqeer (City University of Hong Kong), Dr. Eunice Lau (Queen Elizabeth hospital Hong Kong), and Morayma Temoche (University of California Berkley) for technical assistance. This project is funded by the City University of Hong Kong (grant 9610343, 9667133, and 7200475), the Hong Kong Health, and Medical Research Fund (grant 9211101), the Hong Kong Research Grants Council (grant 21106616), the National Natural Science Foundation of China (grant 81602514, 81773246 and 81770099) and Shenzhen Science and Technology Innovation Fund (grant JCYJ20170413115637100).

## Author contributions

W.M.U. performed the experiments and collected the data with assistance from T.C.P. (EV purification, microscopy, and animal experiments), Y.Y.K. (animal experiments and in vivo imaging), L.T.V. (EV characterization, qRT-PCR, and animal experiments), V.M. (histopathology), B.P. (animal experiments and sucrose gradients), Y.S.C. (FACS and EV characterization), L.W. (cell and EV characterization), S.M.C. (animal experiments), A.A. (electron microscopy), and A.B.-L.H. (leukemia models). M.T.N.L. conceptualized the project, obtained funding, trained the students, designed, and set up the experiments, analyzed the data, and wrote the manuscript with help from J.S. (experimental design, data analysis, funding, and training), W.C.C. (resources and training), N.S.-C. (resources and writing), M.Y. (funding and training), A.Y.H.L. (funding and consultation), and W. M.U. (data analysis and writing).

## Additional information

**Competing interests:** The authors declare no competing interests.

