## [Peer Review File · Nature Communications]

Reviewers' comments:

Reviewer #1 (Remarks to the Author):

In this manuscript, the authors suggest extracellular vesicles derived from red blood cells (RBCs) as delivery vehicles for oligonucleotides and RNAs. They have demonstrated in several cell line models uptake and function, here as impaired miRNA activity by either antagomirs or CRISPR-mediated gene editing. Moreover, they analysed in vivo particle distribution upon intra-tumour and intraperitoneal application and showed in vivo activity with the intratumoural route. Although describing an interesting and novel approach, additional information is required to allow for any meaningful appreciation by the readership.

Comments:

1. The main selling point of this manuscript is the use of RBC-derived vesicles for the in vivo delivery of modulatory oligonucleotides or RNAs. However, although some data regarding a systemic application are provided, all functional data have been obtained using intra-tumour injection. The functional application of such particles in a systemic setting has not been described, but would add significant, if not crucial value to the manuscript. Moreover, it constitutes an absolute requirement for targeting leukemia as implied by the authors.
2. In general, insufficient information is provided regarding the physicochemical characterisation of the particles. What is the PDI and what is the nucleic acid load of these particles? How does electroporation affect size distributions? What is the shelf life? Do these particles aggregate over time? What is their surface charge? Moreover, the authors mention in their final sentence a tolerance of these particles towards multiple freezing-thawing cycles, but do not provide any data supporting this statement.
3. Similarly, several crucial PK parameters are missing. What is their circulation time in the blood stream? What does the signal shown in EDF8 indicate? Does it reflect accumulation in the skin or also presence of particles in deeper tissues?
4. What are the sequences and modifications of the mir125 ASO?
5. How many independent experiments contributed to figures 2D and E?
6. ANOVA might be more appropriate than Student's Ttest in extended data fig. 7.
7. To what extent is the normalisation of in vivo effects by GAPDH and U6b affected by murine transcripts? Are primers used for normalisation human-specific? To what extent do these data truly indicate intracellular changes or simply reflect loss of human cells?
8. Figure 4B does not show Cas9 expression, but cell-associated Cas9 transcript levels.
9. Does figure 4F show total or non-mutated levels of mir125b?
10. Minor point: P7, I13: acute myeloid leukemia, not acute amyloid leukemia.
11. Minor point: P7, I20: RPMI1640

Reviewer #2 (Remarks to the Author):

Waqas et al showcase a strategy to deliver ASOs and mRNA using RBC derived EVs. The concept of using RBC-derived EVs for drug delivery is novel. However, there are certain aspects, such as lack of important controls in some experiments and lack of purification steps after loading that prohibit the work from being published in Nature Communications.

For the future submission, the author should consider the following:

1. The authors have misunderstood the concept of anti-miR, the mechanism is through steric blocking (or in certain cases RNaseH mediated cleavage) and not through RNAi.
2. Figure 1e., extended Figure 2. The authors should mention in the figure legend or in the result

section, what EVs dose was used in MOML13 uptake experiments.

3. Figure 1D. Authors should include in total 3 EV markers, and one EV contamination marker such as GM130 or calnexin. In addition, the authors should have a much better resolution of TEM images of EVs, and should also include zoomed out TEM images of EVs.

4. In extended figure 3. Authors described different electroporation techniques, but they did not describe what voltage was used to electroporate EVs alone without dextran. In addition, dextran can bind to latex beads and therefore they should include a control reaction with 250V with dextran only.

5. Overall, the authors have verified the loading efficiency using Flow cytometry. However, this method is semi-quantitative and a quantitative method such as FCS or NTA with fluorescent tags. Also, an extra purification step should be included to remove excess cargo.

6. Ideally, the authors should perform a sucrose or optiprep gradient on electroporated EVs to show that the cargo is specifically enriched in EV fraction.

7. For mRNA loading experiments, the authors should perform qPCR for detecting levels of cas9 mRNA in purified EVs post electroporation and in addition should include an RnaseH digestion step in order to rule out the possibility that mRNA is on the outside of the EVs.

8. In Extended Fig, 2. authors nicely show uptake of PKH26 labeled EVs, however, the authors should also include a control experiment using heparin to block EV uptake and thus confirm uptake of EVs and not the mere transfer of PKH26 dye.

9. In Extended figure 7 A, the authors should correct the legend. Now it states that only one dose of EVs was used, however, multiple doses of EVs were used in the experiment.

10. In Extended figure 7 b, authors should include standard deviation in the graphs..

11. In Figure 3b, authors have shown very nicely the in vivo uptake of PKH26 labelled EVs, but the author should also include a crucial control group; PBS only labelled the same was as an EV sample. This will help in removing the signal from the free dye in the EV preparation.

12. It should be clearly stated throughout the manuscript what doses have been used in the different experiments. This is not clear.

13. Figure 3g; Authors clearly show therapeutic effect with miR125b anti-miR loaded EVs, however, they need to include a crucial control group, anti-miR 125b alone. Otherwise it is not possible to say that this is EV-mediated delivery.

14. In figure 4c. the authors show through immunofluorescence presence of HA Cas9 in the cells, however, the authors should repeat this experiment, as the cell densities among different groups are varying, especially in the treatment group. In addition, the authors should include a control group where only Cas9 mRNA is electroporated alone without EVs.

15. They can also show the presence of Cas9 protein in recipient cells by performing western blot and staining against anti Cas9 or anti HA tag.

16. Overall the authors need to justify why there was no purification step after the electroporation step to remove un-encapsulated cargo in all the experiments performed in this manuscript.

17. Authors mention in discussion section in line 34 page 6, that RNAs are stable in RBCEVs, but they have not performed any experiment to showcase if different RNA-based cargos are stable in EVs or not.

18. Authors need to list the sequences of various ASO used in this study and need to define various modifications used.

19. Finally, if the authors can provide the particle numbers instead of EVs protein amounts used, this would aid in understanding loading efficiency and dosing in various experiments for the readers.

Dear reviewers,

Thank you very much for the thoughtful and constructive comments on our manuscript “Efficient RNA drug delivery using red blood cell extracellular vesicles (RBCEVs)”. We have done all the experiments and made all the modifications you suggested. Below please find our point-by-point response to your comments.

Responses to Reviewer #1:

1. The main selling point of this manuscript is the use of RBC-derived vesicles for the in vivo delivery of modulatory oligonucleotides or RNAs. However, although some data regarding a systemic application are provided, all functional data have been obtained using intra-tumour injection. The functional application of such particles in a systemic setting has not been described, but would add significant, if not crucial value to the manuscript. Moreover, it constitutes an absolute requirement for targeting leukemia as implied by the authors.

Following the reviewer’s suggestions, we have established a leukemia xenograft model in collaboration with Dr. Anskar Leung, a well-known expert in leukemia research. Using this model, we have successfully demonstrated a therapeutic effect in leukemia, by systemically administering 125b-ASO-loaded RBCEVs (Fig. 4).

2. In general, insufficient information is provided regarding the physicochemical characterization of the particles. What is the PDI and what is the nucleic acid load of these particles? How does electroporation affect size distributions? What is the shelf life? Do these particles aggregate over time? What is their surface charge? Moreover, the authors mention in their final sentence a tolerance of these particles towards multiple freezing-thawing cycles, but do not provide any data supporting this statement.

We have performed a number of experiments to better characterize the physicochemical properties of RBCEVs:

- The polydispersity (PDI) and surface charge (zeta potential) of the EVs are shown in Extended data Fig. 1 b-c.
- We have performed several experiments to determine the nucleic acid load. In Extended Data Fig. 6d, we incubated FAM-ASO electroporated EVs or an unelectroporated mixture of FAM-ASO and RBCEVs with 50% FBS that contained various nucleases at 37°C for 72 hours. Fluorescent signals declined at a much lower rate in electroporated EVs than unelectroporated EVs. Approximately 62% of FAM-ASO were loaded into the electroporated EVs. Similarly, in Fig. 5b, we compared the stability of Cas9 mRNA associated with RBCEVs, with or without electroporation, after incubation with RNase. Approximately 18% of Cas9 mRNA was protected from RNase by the RBCEVs after electroporation.

- As shown in Extended Data Fig. 6c, electroporation did not affect the size distribution of RBCEVs in fraction 6 (most enriched for EVs) of the sucrose gradient.
- We routinely store the EVs at -80C and do not find any change in the EV concentration after thawing, even after a year of freezing. However, we do not know the maximal shelf life of RBCEVs.
- Please find the TEM images and concentration measurements of RBCEVs after multiple freeze-thaw cycles in Extended Data Fig. 2. There was no increase in aggregation or any significant change in the size and morphology of the EVs. Hence, the quality of the EVs was not affected by freeze-thaw cycles.
- As mentioned, we did not observe any increase in aggregation of the EVs after multiple freeze-thaw cycles. In general, there was little aggregation of the EVs after the purification, as shown by the TEM and Nanosight data. Occasionally, we found some batches of RBCEVs with more aggregation due to unknown problems associated with the donors. There was also a significant increase in aggregation after electroporation. These aggregates were often soluble after pipetting.

3. Similarly, several crucial PK parameters are missing. What is their circulation time in the blood stream? What does the signal shown in EDF8 indicate? Does it reflect accumulation in the skin or also presence of particles in deeper tissues?

- We have determined the circulation time of PKH26-labeled RBCEVs in the blood stream of NSG mice as shown in Fig. 4a-b. This time was approximately 6 hours.
- We are not sure if the PKH26 signal previously shown in Extended Data Fig. 8b indicated the accumulation of the signals in the organs, the skin or the deeper tissues because we could not distinguish PKH26 signals from the autofluorescence very well. Therefore, we repeated the experiment using DiR, an infrared dye that is clearly distinguishable from the background autofluorescence. The DiR images showed a clear correlation between the in vivo and ex vivo fluorescent images of the organs (now in Extended Data Fig. 11b-c). There was no signal from the skin, but some signal in the lung, liver, spleen, kidneys, stomach and intestines. However, we could not image DiR fluorescence on the cryosection slides as our (and most) microscopes are not able to reach the infrared emission (780 nm), hence we still present the PKH26 confocal images in Extended Data Fig. 11d as additional evidence for the in vivo uptake of RBCEVs.

4. What are the sequences and modifications of the mir125 ASO?

The sequence of miR-125b ASO is “UCACAAGUUAGGGUCUCAGGGA” with a 2’O-methyl modification at every ribonucleotide.

5. How many independent experiments contributed to figures 2D and E?

We performed each of these experiments twice: the first time with one plate of cells and the second time with 2 plates of cells from 2 different passages. We consider the three cell passages as three biological replicates. Since there was one condition missing in the last replicate of Fig. 2D, we repeated the experiment once more and plotted the complete data in this panel with three independent repeats.

6. ANOVA might be more appropriate than Student’s Ttest in extended data fig. 7.

This is a good suggestion. We have replaced the t-test result with ANOVA test results in Extended Data Fig. 7 (renamed as Extended Data Fig. 10)

7. To what extent is the normalisation of in vivo effects by GAPDH and U6b affected by murine transcripts? Are primers used for normalisation human-specific? To what extent do these data truly indicate intracellular changes or simply reflect loss of human cells?

The qPCR analysis of miR-125b relative to U6b in the CA1a tumors is not specific to human cells. Both miR-125b and U6b RNAs are 100% conserved in humans and mice, hence the primers bind to the RNAs of both tumor cells and stromal cells in the tumor lysates. We do not exclude the possibility that RBCEVs were taken up by the mouse stromal cells in addition to the human tumor cells, and the ASO might suppress both human and mouse miR-125b. Thus the data indicate an intracellular knockdown of miR-125b, but not the loss of human cells.

8. Figure 4B does not show Cas9 expression, but cell-associated Cas9 transcript levels.

We have replaced the term “Cas9 expression” with “Cas9 mRNA level”.

9. Does figure 4F show total or non-mutated levels of mir125b?

This figure shows the level of wild-type miR-125b as the Taqman primers and probe are very specific to the sequence of the wild-type mature miR-125b. Most of the mutations that we found affect the sequence of mature miR-125b. Hence, the qPCR should not detect these mutants.

10. Minor point: P7, I13: acute myeloid leukemia, not acute amyloid leukemia.

We have corrected the typo.

11. Minor point: P7, I20: RPMI1640

We have added “1640”.

Responses to Reviewer #2:

1. The authors have misunderstood the concept of anti-miR, the mechanism is through steric blocking (or in certain cases RNaseH mediated cleavage) and not through RNAi.

We have corrected this term in the abstract and the discussion.

2. Figure 1e., extended Figure 2. The authors should mention in the figure legend or in the result section, what EVs dose was used in MOML13 uptake experiments.

We have mentioned the doses of the EVs in the figure legends and the methods.

3. Figure 1D. Authors should include in total 3 EV markers, and one EV contamination marker such as GM130 or calnexin. In addition, the authors should have a much better resolution of TEM images of EVs, and should also include zoomed out TEM images of EVs.

Following the reviewer’s suggestions, we have tested an additional EV marker, stomatin (STOM), and showed that it was highly enriched in RBCEVs (Extended Data Fig. 1d). We also tested calnexin (CANX) and found it to be absent in both RBCs and RBCEVs, as RBCs do not have any endoplasmic reticulum.

We have also obtained high resolution TEM images. Please find a representative image in Fig. 1 (zoomed out) and a few more in Extended Data Fig. 2.

4. In extended figure 3. Authors described different electroporation techniques, but they did not describe what voltage was used to electroporate EVs alone without dextran. In addition, dextran can bind to latex beads and therefore they should include a control reaction with 250V with dextran only.

We used 250V for the mock electroporation of EVs. We knew that Dextran bound to latex beads when we incubated Dextran with the EVs without electroporation (the third condition), therefore we gated the beads based on this control condition (Extended Data Figure 5).

5. Overall, the authors have verified the loading efficiency using Flow cytometry. However, this method is semi-quantitative and a quantitative method such as FCS or NTA with fluorescent tags. Also, an extra purification step should be included to remove excess cargo.

We have further quantified and verified the loading efficiency using a fluorescent spectrometer (plate reader) as shown in Extended Data Fig. 6. We tried to purify the electroporated EVs using ultracentrifugation and found a very tight pellet that could only break into large sticky EV aggregates, even after vigorous pipetting. Ultracentrifugation of the electroporated EVs using sucrose gradients was better for purifying the EVs from unbound cargo (as shown in Extended Data Fig. 6). However, the EVs mixed with high density sucrose were not suitable for most functional assays. We also tried to purify the electroporated EVs using centricon ultrafiltration, but found an unacceptably high rate of 60-70% loss of the EVs. Thus, we could only continue with our current methods.

6. Ideally, the authors should perform a sucrose or optiprep gradient on electroporated EVs to show that the cargo is specifically enriched in EV fraction.

We have performed the sucrose gradient separation of FAM-ASO electroporated EVs, as shown in Extended Data Fig. 6, and found that the ASO was enriched in EV fractions.

7. For mRNA loading experiments, the authors should perform qPCR for detecting levels of cas9 mRNA in purified EVs post electroporation and in addition should include an RnaseH digestion step in order to rule out the possibility that mRNA is on the outside of the EVs.

We have incubated unelectroporated and Cas9-mRNA-electroporated EVs with RNase If (an RNA endonuclease that digests single stranded RNA, made by New England Biolabs) and found that a significant portion of the electroporated Cas9 mRNA was protected against degradation (Fig. 5b).

8. In Extended Fig, 2. authors nicely show uptake of PKH26 labeled EVs, however, the authors should also include a control experiment using heparin to block EV uptake and thus confirm uptake of EVs and not the mere transfer of PKH26 dye.

As suggested, we have repeated the experiment to test heparin and was able to block EV uptake as shown in Extended Data Fig. 4.

9. In Extended figure 7 A, the authors should correct the legend. Now it states that only one dose of EVs was used, however, multiple doses of EVs were used in the experiment.

We have indicated the comparison of multiple doses in the figure legend (renamed as Fig. 10a).

10. In Extended figure 7 b, authors should include standard deviation in the graphs.

We have repeated the experiment and shown the error bars (SEM) in the graph (renamed as Fig. 10b).

11. In Figure 3b, authors have shown very nicely the in vivo uptake of PKH26 labelled EVs, but the author should also include a crucial control group; PBS only labelled the same as an EV sample. This will help in removing the signal from the free dye in the EV preparation.

The labeled EVs were washed extensively using one round of ultracentrifugation with 60% sucrose cushion and two more rounds of ultracentrifugation with PBS. There is a very small amount of unbound PKH26 dye in the EV prep as shown by a 0.7% shift in PKH26 intensity of MOLM13 cells that were incubated with the supernatant of the last wash, as shown in the new Extended Data Fig. 4. Therefore, we used the supernatant control to determine the background fluorescence when gating the PKH26+ population. We also used the supernatant of DiR-labeled EVs as the controls in our new in vivo uptake experiment, shown in Fig. 4d and Extended data Fig. 11.

12. It should be clearly stated throughout the manuscript what doses have been used in the different experiments. This is not clear.

We have included the doses for every experiment in the figure legends and methods.

13. Figure 3g; Authors clearly show therapeutic effect with miR125b anti-miR loaded EVs, however, they need to include a crucial control group, anti-miR 125b alone. Otherwise it is not possible to say that this is EV-mediated delivery.

We have repeated the experiment with NC/125b-ASO alone and included these conditions in Fig. 3g. 125b-ASO did not have any effect on the tumor growth or on miR-125b levels. This is consistent with our data in Fig. 1i that showed no uptake of FAM-ASO by MOLM13 cells without any EVs or transfection reagents.

14. In figure 4c. the authors show through immunofluorescence presence of HA Cas9 in the cells, however, the authors should repeat this experiment, as the cell densities among different groups are varying, especially in the treatment group. In addition, the authors should include a control group where only Cas9 mRNA is electroporated alone without EVs.

We have repeated the experiment with the same seeding density and included a control group in which MOLM13 cells were electroporated with Cas9 mRNA alone using the same condition for RBCEV electroporation (now in Fig. 5d). We did not find any Cas9 positive cells but a lot of dead cells among the electroporated cells (so the density of these cells became lower) hence the electroporation we optimized for RBCEVs was not effective for the cells. We also tried other conditions that were used for electroporation of suspension cells but did not find any successful electroporation of Cas9 mRNA in MOLM13 cells.

15. They can also show the presence of Cas9 protein in recipient cells by performing western blot and staining against anti Cas9 or anti HA tag.

The images that we presented in Fig. 4 (renamed as Fig. 5) are the immunostaining images of Cas9-HA using an anti-HA-tag antibody. This staining is better than a Western blot as it reveals the nuclear localization of the delivered Cas9 protein.

16. Overall the authors need to justify why there was no purification step after the electroporation step to remove un-encapsulated cargo in all the experiments performed in this manuscript.

As mentioned above, we tried to purify the electroporated EVs using ultracentrifugation or ultrafiltration but encountered problems with aggregates and substantial loss of EVs respectively. We found that naked ASOs were not able to enter the cells (Fig. 1i) and administration of ASOs alone had no functional effect in vivo (Fig. 3g-k). Hence, we believe that the unbound ASOs in the electroporated EV preps are unable to interfere with the observed phenotypes. In the future, we will look for a new washing method that allows us to recover a good amount of electroporated EVs.

17. Authors mention in discussion section in line 34 page 6, that RNAs are stable in RBCEVs, but they have not performed any experiment to showcase if different RNA-based cargos are stable in EVs or not.

We have performed the stability tests for FAM-ASO in RBCEVs, after a 72-hour incubation with 50% serum (and thus various RNAses) at 37°C (Extended Data Figure 6d). The data demonstrated that FAM-ASO was significantly more stable in electroporated RBCEVs, than in the un-electroporated RBCEV mixture.

18. Authors need to list the sequences of various ASO used in this study and need to define various modifications used.

We have listed the sequences and modifications of the ASOs and gRNA in the method section.

19. Finally, if the authors can provide the particle numbers instead of EVs protein amounts used, this would aid in understanding loading efficiency and dosing in various experiments for the readers.

We have provided the particle numbers in addition to the amounts of EVs in the figure legends and methods.

We hope you are satisfied with the revision.

Reviewers' comments:

Reviewer #1 (Remarks to the Author):

The authors addressed my comments in points 2-11 to my satisfaction. However, I do have some remaining questions regarding their experiments addressing point 1.

1. The authors show nicely the tissue distribution of RBCEVs in liver, spleen, stomach and intestine. Although they (quite correctly) state that in particular liver and spleen are important for leukaemia, the most relevant organ is the bone marrow. Do the authors have evidence for accumulation of RBCEVs in the bone marrow by e.g. flushing out the femurs?
2. Does the reduced bioluminescence translate into an increased median survival?
3. Figure 4j requires more explanation in the legends. In its current form, it is largely non-instructive for readers without substantial haematological knowledge.

Reviewer #2 (Remarks to the Author):

While the authors have made efforts to address the substantial criticisms of the reviewers, the manuscript as it stands fails to address several fundamental methodological concerns as follows:

1. Figure legends are in many cases inadequate – it is almost impossible to understand the methods used or the data from the description in the legends e.g. no mention of methodological details in legend for Figure 1.
2. Electroporation – unsatisfactory details are provided. The authors did not describe what voltage was used to electroporate EVs alone without dextran. In addition, dextran can bind to latex beads and therefore they should include a control reaction with 250V with dextran only
3. Loading efficiency –unsatisfactory detail and the authors have almost certainly overestimated the loading efficiencies in their work and the current data appears unreliable. Quantitative methods e.g. FCS or NTA with fluorescent tags should be used.
4. In the case of ASO loading into EVs, PCR or hybridisation based methods should be used to accurately quantify ASO copy numbers loaded
5. Figure 1
 - Title should be altered – the figure shows no direct evidence of ASO delivery
 - FAM-ASO methods are unsatisfactory. The authors should use PCR or hybridisation based methods to detect intracellular ASO delivery
6. Figure 2
 - Title should be altered - the figure shows no direct evidence of ASO delivery
 - 2b why is quantification vs U6B – were other standards evaluated?
 - 2b – no statistical clarity in relation to this figure – unclear what P value refers to
 - 2b Experiment is flawed – the dose response should relate to EV particle numbers not EV mass
 - 2b no control with ASO alone or ASO with transfection reagent (the ASO alone or if 2'OMe chemistry should internalise in cells via gymnotic delivery over time)
 - 2b how many times was this experiment performed – n numbers and biological and technical replicate data not clear
7. Figure 5
 - 2c data unclear. Why Cas9 quantified relative to GAPDH? How many other controls were evaluated to find the most appropriate comparator
 - 2c – what do the 3, 6, 12 pmol refer to? EV dose response should be performed in particle numbers
 - For Cas9 mRNA loading experiments, the authors should perform qPCR for detecting levels of cas9 mRNA in purified EVs post electroporation.
 - Cas9 protein delivery to cells should be demonstrated by quantified western blot

8. Finally, the authors have failed to adequately justify why there was no purification step after the electroporation step to remove un-encapsulated cargo in all the experiments performed in this manuscript.

Dear reviewers,

Thank you very much for providing more thoughtful and constructive comments on our manuscript “Efficient RNA drug delivery using red blood cell extracellular vesicles (RBCEVs)”. We have revised the manuscript again according to your valuable suggestions. Below please find our point-by-point responses to your comments.

Responses to Reviewer #1:

1. The authors show nicely the tissue distribution of RBCEVs in liver, spleen, stomach and intestine. Although they (quite correctly) state that in particular liver and spleen are important for leukaemia, the most relevant organ is the bone marrow. Do the authors have evidence for accumulation of RBCEVs in the bone marrow by e.g. flushing out the femurs?

We agree with the reviewer that the bone marrow is the primary organ that leukemia cells home into. Due to the blocking of DiR signals by the dense bone, we were not able to detect DiR from the bone marrow, using the in vivo imaging system. DiR fluorescence was not detectable using our microscopes or flow cytometer due to its long excitation/emission wavelength (750/780 nm). Therefore, we repeated the experiment using RBCEVs labeled with Vivo-Track-680 (VVT), a near-infrared membrane dye that is detectable by FACS (Fig. 6f). We isolated bone marrow cells from the leukemic mice by flushing the femurs as the reviewer suggested. Indeed, we found VVT fluorescence in nearly 40% of the bone marrow cells from the NSG mice injected i.p. with VVT-labeled RBCEVs using FACS analysis (Fig. 6g-h). Hence, RBCEVs were taken up robustly by bone marrow cells.

2. Does the reduced bioluminescence translate into an increased median survival?

While we think this is a very good question, it is technically impossible to study the survival effect because our IACUC dictates that we have to sacrifice the mice when they show any signs of distress and weakness or when the tumors are larger than 1.8 cm in diameter, according to our animal protocol. In our study, the leukemic mice were still alive when they showed very clear distress and weakness, including reduced activity, piloerection, partially closed eyelids and sometimes paralysis. In the CA1a breast cancer model, most of the mice also survived when the tumors reached 1.8 cm in diameter and some of them showed clear signs of pain and distress at this stage. We cannot keep the animals beyond these points, to examine the differences in survival, as it is unethical to do so.

3. Figure 4j requires more explanation in the legends. In its current form, it is largely non-instructive for readers without substantial haematological knowledge.

We have revised the legend of Fig. 4j which is now Fig. 7f. We used arrows to indicate clusters of

infiltrating leukemia cells that have larger nuclei than normal cells.

Responses to Reviewer #2:

While the authors have made efforts to address the substantial criticisms of the reviewers, the manuscript as it stands fails to address several fundamental methodological concerns as follows:

1. Figure legends are in many cases inadequate – it is almost impossible to understand the methods used or the data from the description in the legends e.g. no mention of methodological details in legend for Figure 1.

We have revised all the legends to explain the experiment better. However, we cannot include all the methodological details in the legends, due to space limits and redundancy, since all the technical information are already provided in the methods section. We have divided the data in figure 1 into 2 figures (currently Fig. 1 and 2 with some additional data) to create more space for the legends, including necessary information about the methods.

2. Electroporation – unsatisfactory details are provided. The authors did not describe what voltage was used to electroporate EVs alone without dextran. In addition, dextran can bind to latex beads and therefore they should include a control reaction with 250V with dextran only

We have described in the figure the voltage used to electroporate EVs alone without dextran (250V). In this revision we also provided the control reaction with dextran only, electroporated at 250V (Extended Data Fig. 4b). Electroporation did increase the binding of Dextran to the beads, but to a lesser extent than the binding of Dextran-electroporated EVs.

3. Loading efficiency – unsatisfactory detail and the authors have almost certainly overestimated the loading efficiencies in their work and the current data appears unreliable. Quantitative methods e.g. FCS or NTA with fluorescent tags should be used.

If we understood correctly, the reviewer referred to fluorescent correlation spectrometry and nanoparticle tracking analysis as FCS and NTA. FCS is an advanced microscopy method for quantification of fluorescence at the single-molecule level. It was first described for the detection of fluorescent labeled EVs by Wyss et al (Anal. Chem., 2014). We are very interested in this method. However, it requires an expensive upgrade of our department's confocal microscope that we cannot afford in the short-term. We tried to use NTA instead. With our Nanosight NS300, one of the latest models of NTA, however, we were not able to detect FAM fluorescence from the FAM-ASO-loaded EVs. We discussed with a Nanosight specialist and found that NTA is usually used for EVs with fluorescent labels on their membrane, but it is not sensitive enough to detect fluorescent-labeled cargos inside the EVs.

We agree with the reviewer that the ASO loading efficiency was overestimated by the fluorescent intensity in the serum-treatment experiment probably due to the retention of the fluorescent dye after the RNA was degraded. Hence, we have attempted to quantify the ASO directly. We first used 10% native gels to separate the unbound ASO from electroporated EVs and visualized the ASO using SYBR Gold. We found ~76% ASOs migrated into the gel from the ASO-electroporated RBCEV sample relative to

the untreated ASOs (Fig. 2c). Hence, ~24% of the ASOs were loaded into the RBCEVs by electroporation. Similar loading efficiency was observed with FAM fluorescence from the RBCEVs electroporated with FAM-ASO (Extended Data Fig. 6a). It is noteworthy that unbound ASO appeared as a single band in all samples including the electroporated RBCEVs and no FAM signal was detected in the well. Hence, the electroporation did not cause any aggregation of the ASO, unlike the aggregation of Cy3 or Cy5-labeled oligonucleotides that we and others have observed before.

To test if the ASOs were contained within the EVs, we treated the electroporated RBCEVs with RNase-If that led to a degradation of ~80% 125b-ASO, relative to the untreated ASO quantified using a sequence-specific Taqman qRT-PCR; whereas, the same amount of ASOs in an unelectroporated mixture with RBCEVs was completely degraded (Fig. 3b). This data suggests that approximately 20% of the ASOs were loaded into RBCEVs by electroporation and thus, protected from the RNase. This proportion of loaded ASOs is 4% lower than what we observed with the native gel, probably because 4% of the ASOs remained on the EV membrane after electroporation that was not separated by the native gel but digested by the RNase.

We have also done a direct quantification of the Cas9 mRNA after RNase treatment of RBCEVs electroporated with Cas9 mRNA and found 18% Cas9 mRNA loaded into RBCEVs (Fig. 8b).

4. In the case of ASO loading into EVs, PCR or hybridisation based methods should be used to accurately quantify ASO copy numbers loaded

To quantify the exact copy number of 125b-ASO, we generated a standard curve of the ASO amplification using Taqman qRT-PCR (Extended Data Fig. 6c). Based on the sequence-specific Taqman qRT-PCR of RNase-treated ASOs and RBCEVs (Fig. 3b), the amount of 125b-ASO loaded into 6.2×10^{11} RBCEVs was approximately 24×10^{12} copies in total, which is ~38.7 copies per EV on average. Furthermore, we found $\sim 21 \times 10^9$ copies of 125b-ASO in MOLM13 cells after a 72-hour-incubation with 12×10^{11} 125b-ASO loaded RBCEVs (Fig. 3c).

5. Figure 1

- Title should be altered – the figure shows no direct evidence of ASO delivery
- FAM-ASO methods are unsatisfactory. The authors should use PCR or hybridisation based methods to detect intracellular ASO delivery

We have divided figure 1 into Fig. 1 and 2 with new titles. As mentioned above, we have shown a direct quantification of the ASO based on the native gel and Taqman qRT-PCR (Fig. 2c and Fig. 3b-c).

6. Figure 2

- Title should be altered - the figure shows no direct evidence of ASO delivery
- 2b why is quantification vs U6B – were other standards evaluated?

We have provided the direct evidence of the ASO delivery as mentioned above, hence we would like to keep the title of this figure (renumbered Fig. 3) as before.

U6b is the most commonly used internal control for miRNA Taqman qPCR (Thermo Fisher Scientific) as U6b RNA is a housekeeping small RNA that has a similar length as miRNAs. Hence, it can be detected using the same RT-PCR method. To confirm the inhibition of miR-125b in MOLM13 cells treated with

125b-ASO loaded RBCEVs, we quantified miR-125b using the miRCURY-LNA Universal RT miRNA PCR, a miRNA qRT-PCR kit from Exiqon that is totally different from the Taqman miRNA qRT-PCR. We normalized miR-125b to miR-103a, an internal control recommended by Exiqon, due to its consistent expression in different conditions tested before. To facilitate the comparison of the 2 methods, we plotted miR-125b fold change relative to the untreated sample. Quantification of miR-125b relative to miR-103a using miRCURY-LNA qRT-PCR confirmed the knockdown of miR-125b by the RBCEV-delivered ASO in a similar pattern as what we observed using the Taqman qRT-PCR (Extended Data Fig. 10a vs Fig. 3d).

- 2b - no statistical clarity in relation to this figure - unclear what P value refers to

The P value in Fig. 2b (currently Fig. 3d) refers to one-way ANOVA test as mentioned in the legend.

- 2b Experiment is flawed - the dose response should relate to EV particle numbers not EV mass

We have replaced the mass with the particle numbers in this figure and all other figures, just as the reviewer suggested.

- 2b no control with ASO alone or ASO with transfection reagent (the ASO alone or if 2' OMe chemistry should internalise in cells via gymnotic delivery over time)

We have provided the qPCR data with ASO alone condition (Fig. 3e-f). The ASO alone did not have any effect on miR-125b or BAK1 expression.

- 2b how many times was this experiment performed - n numbers and biological and technical replicate data not clear

N= 3 cell passages as mentioned in the legend. Each experiment was repeated in 3 different cell passages that were considered as 3 biological replicates. Each value is an average of 6 qPCR readings as 2 technical replicates was performed for each RNA sample from each cell passage.

7. Figure 5

- 2c data unclear. Why Cas9 quantified relative to GAPDH? How many other controls were evaluated to find the most appropriate comparator

GAPDH is a common internal control for SYBR Green qPCR of mRNA. As the reviewer requested, we repeated the qPCR of *Cas9* mRNA using *GAPDH*, *18S* rRNA and beta-actin (*ACTB*) as normalizers. To compare the results, we calculated the fold change of the *Cas9* mRNA in all conditions relative to the 3 pmol *Cas9* mRNA treatment. The fold change of *Cas9* mRNA relative to *GAPDH* was very similar to that relative to *18S* rRNA and *ACTB* (Fig. 8c and Extended data Fig. 14b).

- 2c - what do the 3, 6, 12 pmol refer to? EV dose response should be performed in particle numbers

It is the amount of Cas9 mRNA used for electroporation, as mentioned in the legend.

- For Cas9 mRNA loading experiments, the authors should perform qPCR for detecting levels of cas9 mRNA in purified EVs post electroporation.

We have shown this in Fig. 8b (previously Fig. 5b) condition 4 and 5: Cas9 mRNA in electroporated EVs with and without RNase treatment.

- Cas9 protein delivery to cells should be demonstrated by quantified western blot

We have provided a quantified Western blot of Cas9 protein in Fig. 8f.

8. Finally, the authors have failed to adequately justify why there was no purification step after the electroporation step to remove un - encapsulated cargo in all the experiments performed in this manuscript.

This is a fair suggestion on its own. But in the context of our previous experiments it is unnecessary since we have already shown that the ASO is not taken up by the cells without RBCEVs or other carriers (Fig. 2d).

The ASOs that we used are RNA oligonucleotides with 2' O-methyl modifications that enhance the ASO stability but do not facilitate cellular uptake. We have included the ASO-only control in all our breast cancer treatment and showed that administration of the 125b-ASO alone had no effect on tumor growth, invasion, metastasis and miR-125b expression in the tumor compared to the controls while 125b-ASO-loaded RBCEVs suppressed tumor growth, invasion and metastasis significantly by inhibition of miR-125b in the tumor (Fig. 5a-g).

In this revision, we also provided the ASO-only control in the in vitro treatment of leukemia MOLM13 cells with 125b-ASO-loaded RBCEVs experiments, just as the reviewer suggested. Incubation of the cells with 125b-ASO had no effect on the expression of miR-125b and *BAK1*, while the treatment with 125b-ASO-loaded RBCEVs suppressed miR-125b expression and upregulated *BAK1* level significantly (Fig. 3d-e). These data suggest that unbound ASOs in the electroporated EV preps were unlikely to interfere with the observed phenotypes. Therefore, it was unnecessary to wash the electroporated EVs in our experiments.

In addition, when we tried to wash the electroporated EVs using ultracentrifugation or ultrafiltration, we encountered problems with aggregates and 50-70% loss of EVs respectively. Hence it is not technically feasible to repeat every experiment in this manuscript with such washing steps. In the future, we will look for a new washing method that allows us to recover a good amount of electroporated EVs so that we can provide purified electroporated EVs for clinical applications.

We hope you are reasonably satisfied with our revisions. Thank you.

Reviewers' comments:

Reviewer #1 (Remarks to the Author):

The authors addressed my comments and questions to my satisfaction. I have two minor comments:

1. The treatment of MOLM13-transplanted animals did not lead to an increased overall survival. Killing an animal due to weakness or tumour size constitutes a leukaemia-associated death and must not be censored, nor does it invalidate the survival analysis. The authors should state this clearly in the main text, when describing figure 7.
2. The authors should also mention in the legend that the left column of fig 7f shows normal tissue from non-transplanted animals.

Reviewer #2 (Remarks to the Author):

1. Abstract

The authors make the unsubstantiated claim in the abstract that 'current cellular sources for EV production are limited in scalability'. This is not true and is in no way a justification for the current study. At least 2 if not more companies (Codiak, Evox etc) are producing EVs for clinical application without the need for red blood cell derived EVs. There are numerous sources of EVs for suitable clinical trial production. The introductory references to this point should also be removed/clarified

2. Introduction

It is disingenuous of the authors not to acknowledge the substantial body of work in EV based RNA delivery that precedes the current study and of which the current work is simply a derivative. The key papers (Validi et al from 2007, and Alvarez-Erviti et al from 2011) are from almost a decade ago and should be clearly cited as intellectual precedents in the introduction.

3. The criticism in relation to experiments failing to use EV particle numbers for dose response has not been adequately answered. It is not possible to convert EV mass into particle numbers as the authors seem to suggest. The experiments need to be repeated as particle numbers vary in each experiment and with each EV purification and preparation.

4. The authors seem to suggest that 2'O-methyl modified ASOs are not internalised into cells (since we have already shown that the ASO is not taken up by the cells without RBCEVs or other carriers). These experiments should be repeated since the authors will find, as indicated previously that even cells somewhat refractory to ASO uptake (e.g. neurons or myocytes) will internalise 2'O-methyl modified ASOs via gymnotic delivery over time

Dear reviewers,

Thank you very much for providing more thoughtful and constructive comments on our manuscript “Efficient RNA drug delivery using red blood cell extracellular vesicles (RBCEVs)”. We have revised the manuscript again according to your valuable suggestions. Below please find our point-by-point responses to your comments.

Responses to Reviewer #1:

1. The treatment of MOLM13-transplanted animals did not lead to an increased overall survival. Killing an animal due to weakness or tumour size constitutes a leukaemia-associated death and must not be censored, nor does it invalidate the survival analysis. The authors should state this clearly in the main text, when describing figure 7.

As the reviewer suggested, we have included a statement in the main text for figure 7 that “we could not assess the effect of the treatments on the overall survival of the mice due to restrictions defined by our institutional ethics committee. All the mice were sacrificed on day 9 except for 2 control mice that died on day 8”. We killed the mice at the same time to compare the leukemia burden although the control mice were weak and the treated mice were still active at that time. Therefore, we do not wish to and did not claim that our treatment can increase overall survival of cancer-afflicted animals. Instead, our analysis focused on the significant difference in the leukemia burden between the treated and the control group.

2. The authors should also mention in the legend that the left column of fig 7f shows normal tissue from non-transplanted animals.

This is a good suggestion. We have edited Fig. 7f legend accordingly.

Responses to Reviewer #2:

1. Abstract

The authors make the unsubstantiated claim in the abstract that 'current cellular sources for EV production are limited in scalability'. This is not true and is in no way a justification for the current study. At least 2 if not more companies (Codiak, Evox etc) are producing EVs for clinical application without the need for red blood cell derived EVs. There are numerous sources of EVs for suitable clinical trial production. The introductory references to this point should also be removed/clarified

We acknowledge this point from the reviewer, and have toned down our claims on “scalability” in the abstract and main text, as the reviewers suggested. This is because, while it is still difficult in the lab

setting as far as we are aware of, large-scale EV production is indeed possible from other cell sources with industrial-level investments in a company setting.

2. Introduction

It is disingenuous of the authors not to acknowledge the substantial body of work in EV based RNA delivery that precedes the current study and of which the current work is simply a derivative. The key papers (Validi et al from 2007, and Alvarez-Erviti et al from 2011) are from almost a decade ago and should be clearly cited as intellectual precedents in the introduction.

We apologize for the short introduction and for our oversight. As the reviewer suggested, we have clarified the introduction text to highlight the discoveries of endogenous RNA delivery and EV-based RNA, by citing Validi et al and Alvarez-Erviti et al.

3. The criticism in relation to experiments failing to use EV particle numbers for dose response has not been adequately answered. It is not possible to convert EV mass into particle numbers as the authors seem to suggest. The experiments need to be repeated as particle numbers vary in each experiment and with each EV purification and preparation.

We did measure the particle numbers in each EV prep. We have calculated the particle number again and repeated the experiment once more, to make sure that the dose-response effect is reproducible. We now present in the graphs the exact average particle numbers that we used for each dose (Fig. 3d-e, Extended Data Fig. 11 a-c and Extended Data Fig. 15a).

4. The authors seem to suggest that 2' O - methyl modified ASOs are not internalised into cells (since we have already shown that the ASO is not taken up by the cells without RBCEVs or other carriers). These experiments should be repeated since the authors will find, as indicated previously that even cells somewhat refractory to ASO uptake (e.g. neurons or myocytes) will internalise 2' O - methyl modified ASOs via gymnotic delivery over time

Indeed, we observed FAM fluorescence in ~0.4% MOLM13 and NOMO1 cells after 24 hours of incubation with unencapsulated FAM ASOs (Fig. 2d-e and Extended Data Fig. 8c). We have repeated the experiment with FAM-ASO uptake in MOLM13 cells for 5 days. In the first 4 days, the FAM signal increased gradually from 0.4 to 2.1% of MOLM13 cells, suggesting that the unencapsulated FAM ASOs were taken up slowly by a tiny population of MOLM13 cells via gymnotic delivery (Extended Data Fig. 9). Afterwards, the percentage of FAM-positive MOLM13 cells did not increase further with FAM ASO treatment. However, in the same period of time, the delivery of FAM ASOs by RBCEVs occurred at a much higher rate, from 75% after 2 days to 100% FAM-positive MOLM13 cells after 4 days of incubation (Extended Data Fig. 9). These data suggested that although 2'O-methyl modified ASOs can be internalized by the cells via gymnotic delivery, the delivery of ASOs was much more robust by RBCEVs.

We hope you are reasonably satisfied with all our revisions. Thank you very much.

REVIEWERS' COMMENTS:

Reviewer #2 (Remarks to the Author):

No further comments